# SHOOT FROM THE HIP: HESSIAN INTERATOMIC POTENTIALS WITHOUT DERIVATIVES

## ABSTRACT

Fundamental tasks in computational chemistry, from transition state search to vibrational analysis, rely on molecular Hessians, which are the second derivatives of the potential energy. Yet, Hessians are computationally expensive to calculate and scale poorly with system size, with both quantum mechanical methods and neural networks. In this work, we demonstrate that Hessians can be predicted directly from a deep learning model, without relying on automatic differentiation or finite differences. We observe that one can construct SE(3)-equivariant, symmetric Hessians from irreducible representations (irrep) features up to degree $l=2$ computed during message passing in graph neural networks. This makes HIP Hessians one to two orders of magnitude faster, more accurate, more memory efficient, easier to train, and enables more favorable scaling with system size. We validate our predictions across a wide range of downstream tasks, demonstrating consistently superior performance for transition state search, accelerated geometry optimization, zero-point energy corrections, and vibrational analysis benchmarks. We open-source the HIP codebase and model weights to enable further development of the direct prediction of Hessians.

## 1 INTRODUCTION

Over the past decades, molecular simulation has become a cornerstone for many tasks in material discovery and molecular design. Beyond energy and forces, chemists, chemical engineers, materials scientists, and physicists frequently rely on Hessians to enable many critical workflows (Qu & Bowman, 2016; Huang et al., 2018). For example, *second-order geometry optimization* accelerates the determination of reliable equilibrium structures. *Transition-state searches* are needed to find reaction pathways and estimate barrier heights and transition rates. This determines whether a reaction is viable and how to tune selectivity, with high industrial relevance for the chemical and pharmaceutical industry. *Vibrational analyses* connect theory to experiment through infrared and Raman spectra and provide zero-point energies, which are essential for ranking isomers and estimating reaction free energies.

Despite their broad utility, Hessian calculations remain a significant computational bottleneck. For a molecule with $N$ atoms, the Hessian is a $3N \times 3N$ matrix, where each entry requires a mixed second derivative of the electronic energy with respect to two nuclear coordinates

$$\mathbf{H}_{IJ}^{\alpha,\beta} = \frac{\partial^2 E}{\partial \mathbf{R}_I^\alpha \, \partial \mathbf{R}_J^\beta} \tag{1}$$

with $I, J = 1, \ldots, N$ indexing atoms, and $\alpha, \beta \in \{x, y, z\}$ indexing the cartesian $x$, $y$ and $z$ components of the atom positions $\mathbf{R}$.

Traditional approaches rely either on analytic second derivatives or on numerical differentiation of gradients, both of which have severe limitations (Jørgensen & Simons, 1983; Handy & Schaefer, 1984; Komornicki & Fitzgerald, 1993). Analytic second derivatives are available only for a limited set of methods, such as Hartree-Fock, DFT, and MP2; whereas, for higher-level correlated wavefunction methods such as CCSD or CASPT2, they are either not yet implemented in most computational packages or computationally prohibitive. Numerical approaches sidestep analytic derivations but can become infeasible even for medium-sized molecules, requiring $O((3N)^2)$ gradient evaluations (Helgaker et al., 2000).

Most computational chemistry calculations are carried out with Kohn-Sham density functional theory

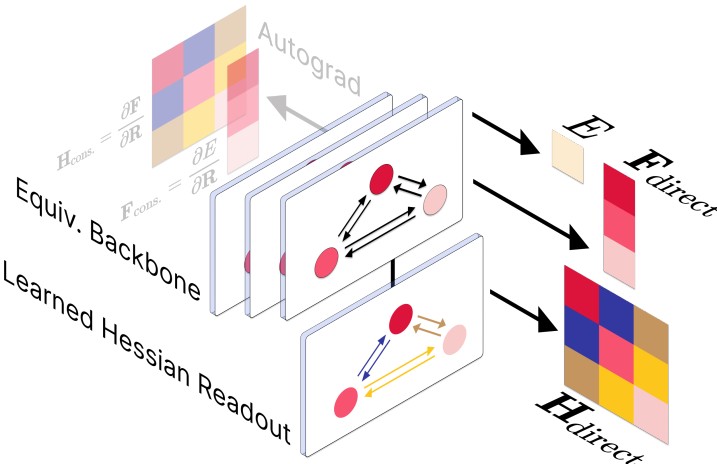

Figure 1: In contrast to previous methods that rely on AD to construct the Hessian matrix, Hessian Interatomic Potentials (HIP) use a learned Hessian readout that predict the Hessian *directly*.

(Kohn, 1965) (KS-DFT), due to its balance of computational cost and accuracy (Bursch et al., 2022). However, the $O(N^4)$ scaling of hybrid KS-DFT with system size remains expensive, which has spurred the development of machine-learning interatomic potentials (MLIPs). MLIPs promise to deliver near-DFT accuracy at a fraction of the cost by incorporating symmetries, thereby drastically reducing the need for extensive training data (Schütt et al., 2018; Thomas et al., 2018; Gasteiger et al., 2021; Schütt et al., 2021; Batatia et al., 2022; Batzner et al., 2022). Recently, some MLIP models have also been adapted to compute Hessians via automatic differentiation (AD). These AD Hessians incur significant computational costs of $O(N^2)$, which presents practical challenges during training and inference (Yuan et al., 2024; Gönnheimer et al., 2025; Rodriguez et al., 2025; Williams et al., 2025; Cui et al., 2025).

In this work we introduce **HIP**: **H**essian **I**nteratomic **P**otentials. HIPs predict the Hessian directly, to eliminate the need for finite differences, coupled-perturbed equation solvers, or automatic differentiation. By building on SE(3)-equivariant neural networks, we demonstrate how HIP can predict Hessians at a fraction of the computational cost of traditional methods. A key observation is that **one can construct the Hessian from spherical harmonic features of the atom nodes and the messages between atoms**, while satisfying equivariance and symmetry constraints by design. This approach is not only faster but also more accurate, enabling a variety of critical tasks in molecular simulations.

Our contributions are the following:

1. We present Hessian Interatomic Potentials (HIP), a method to construct SE(3)-equivariant, symmetric Hessians *directly*, without finite differences, coupled-perturbed solvers, or automatic differentiation, using an equivariant neural network backbone.

2. We show that downstream results can be improved further by using a loss function that targets the relevant Hessian subspace

3. We show that, compared to AD, the predicted Hessians are 10-70x faster for single molecule evaluation and over 70x faster in batched molecular prediction, while also requiring 2-3x less peak memory on small molecules with 5-30 atoms.

4. We show that the predicted Hesians are more accurate on benchmark datasets, with 2x lower Hessian MAE, 3.5x lower eigenvalue MAE, and 1.5x higher eigenvector cosine similarity.

5. Finally, we rigorously validate the practical utility of our predicted Hessians in several downstream applications, such as (i) zero-point energy corrections, (ii) second-order geometry optimization, (iii) transition state search, and (iv) frequency analysis for extrema classification.

## 2 BACKGROUND

**MLIPs and equivariant neural networks** Early works on machine learning interatomic potentials used local atomic environment descriptors in combination with linear regression (Thompson et al., 2015; Shapeev, 2016), Gaussian processes (Bartók et al., 2010), and simple neural networks (Behler & Parrinello, 2007). SchNet (Schütt et al., 2018) was the first work to introduce *rotation-invariant graph neural networks* to predict molecular properties like energies and forces. Next-generation invariant architectures include ViSNet (Wang et al., 2022) and QuinNet (Wang et al., 2024). In parallel, *rotation-equivariant* architectures were developed, such as Cormorant (Anderson et al., 2019), DimeNet (Gasteiger et al., 2020), GemNet (Gasteiger et al., 2021), PaiNN (Schütt et al., 2021), NequIP (Batzner et al., 2022), SphereNet (Liu et al., 2022), MACE (Batatia et al., 2022) and the Equiformer family (Liao & Smidt, 2023; Liao et al., 2024). *Graph neural networks* represent a molecule as a graph $\mathcal{G} = (\mathcal{V}, \mathcal{E})$ in 3D Euclidean space $\mathbb{R}^3$, with nodes/atoms $I \in \mathcal{V}$ at positions $\mathbf{r}_I \in \mathbb{R}^3$ and edges $(I, J) \in \mathcal{E}$ defined by a cutoff radius. At layer $t$, each node carries a feature $\mathbf{h}_I^{(t)}$ that is a direct sum of irreducible SO(3) representations (irreps): $\mathbf{h}_I^{(t)} = \bigoplus_{l=0}^{l_{\max}} \mathbf{h}_I^{(t,l)}$, where $\mathbf{h}_I^{(t,l)} = \{\mathbf{h}_{I,l,m}^{(t)}\}_{m=-l}^{l}$ has $2l + 1$ components. To update the node's features, the model sends messages between each connected node, which are then aggregated in a permutation invariant fashion (usually the sum, or attention-weighted-sum operation) and then added to the receiving node. In SO(3)-equivariant GNNs, node features transform under a global rotation $\mathbf{Q}$ via Wigner $\mathbf{D}$-matrices:

$$\mathbf{h}_{i,l,m}^{(t)}\left(\mathbf{Q}\{\mathbf{r}_k\}_{k=1}^N\right) = \sum_{m'=-l}^{l} \mathbf{D}_{mm'}^{(l)}(\mathbf{Q})\, \mathbf{h}_{i,l,m'}^{(t)}\left(\{\mathbf{r}_k\}_{k=1}^N\right), \tag{2}$$

Translation equivariance is enforced by building messages from relative displacements $\mathbf{r}_{I,J} = \mathbf{r}_J - \mathbf{r}_I$. If we want O(3) instead of SO(3) equivariance, each order-$l$ feature has an extra parity label that determines the transformation under a global coordinate inversion.

Two irreps $\mathbf{p}_{l_1}$ and $\mathbf{g}_{l_2}$ interact using the Clebsch-Gordan tensor product (Thomas et al., 2018)

$$\mathbf{h}_{l_3,m_3} = \left(\mathbf{p}_{l_1,m_1} \otimes_{l_1,l_2}^{l_3} \mathbf{g}_{l_2,m_2}\right)_{m_3} = \sum_{m_1=-l_1}^{l_1} \sum_{m_2=-l_2}^{l_2} \mathbf{C}_{(l_1,m_1),(l_2,m_2)}^{(l_3,m_3)} \mathbf{p}_{l_1,m_1} \mathbf{g}_{l_2,m_2} \tag{3}$$

A message from source atom $J$ to target atom $I$ is then constructed as

$$\mathbf{v}_{I,J,l_3} = \mathbf{v}_{l_3}(\mathbf{h}_I, \mathbf{h}_J, \mathbf{r}_{I,J}) = \sum_{l_1,l_2} w_{l_1,l_2,l_3}(||\mathbf{r}_{I,J}||) \left(\mathbf{f}_{l_1}(\mathbf{h}_I, \mathbf{h}_J) \otimes_{l_1,l_2}^{l_3} \mathbf{Y}_{l_2}\left(\frac{\mathbf{r}_{I,J}}{||\mathbf{r}_{I,J}||}\right)\right) \tag{4}$$

where $\mathbf{Y}_{l_2}$ is the $2l_2 + 1$ dimensional vector of the spherical harmonics of degree $l_2$, $w_{l_1,l_2,l_3} : \mathbb{R} \to \mathbb{R}$ is a learned weighting function, and $\mathbf{f}_{l_1}$ a simple function that takes both node features in, for example concatenation in the case of EquiformerV2. A more efficient variation of equation 4 relying on the reduction of the $SO(3)$ tensor product to the $SO(2)$ tensor products was proposed by eSCN (Passaro & Zitnick, 2023) and adopted by subsequent works, such as EquiformerV2. Finally, the messages are pooled to update the node features, which can be done, for example, via a sum or using graph attention (Liao et al., 2024).

After several layers of message passing, task-dependent readout heads map the node features to the desired targets while respecting symmetry. For example, energies $E$ are SO(3)-invariant per-graph scalars. The energy readout head therefore uses a global pooling and a reduction operation to output a $l=0$ feature. Forces $\mathbf{F}$ are SO(3) equivariant per-atom vectors ($l = 1$), and can be outputted either by a direct force readout head, or as the derivative of the energy $\mathbf{F} = -\nabla_\mathbf{R} E$. As we will see in section 3, Hessians are SO(3)-equivariant per-graph matrices composed of per-atom-pair $3 \times 3$ equivariant sub-matrices.

**Equivariant matrix prediction** Similar to the Hessian, there are other matrices of interest that are rotation-equivariant and that previous work tried to directly regress. For example, PhisNet Unke et al. (2021), QHNet Yu et al. (2023), and WANet Li et al. (2025) try to predict the Fock matrix, and Graph2Mat Febrer et al. (2025) predicts the density matrix.

### 2.1 TRADITIONAL METHODS FOR CALCULATING HESSIANS

**Hessians from DFT** The gold standard for computing nuclear Hessians is to derive them analytically. In Kohn-Sham DFT, this is achieved using the Coupled Perturbed Kohn–Sham (CPKS) method

(Deglmann et al., 2002), which is employed to calculate the ground truth in the dataset and in the workflows of our experiments. If available, analytical Hessians provide the most efficient ab-initio way to obtain Hessians, but they still scale as $O(N^5)$. See Appendix A.1.1 for details. A major downside of the CPKS approach to Hessians is its highly complex implementation and need for exchange-correlation-functional-specific derivations, which makes it not always available. An alternative is to use a finite-difference scheme, requiring only energies or forces. Finite differences also scale $O(N^5)$ for hybrid DFT, although with more numerical noise and a much larger computational prefactor. We provide more details in A.1.2.

**Hessians from MLIPs** MLIPs have gained widespread popularity for their ability to accurately approximate molecular energies and forces. The forces can either be directly predicted (Gasteiger et al., 2021; Liao et al., 2024), or calculated as the derivative of the MLIP energy using automatic differentiation. Previous work has shown that the Hessian can be calculated using AD to some success (Yang et al., 2024; Yuan et al., 2024; Gönnheimer et al., 2025; Rodriguez et al., 2025; Williams et al., 2025; Cui et al., 2025). Although AD Hessians are much cheaper to calculate compared to DFT, they still have some downsides. First, accurately modeling the energy does not guarantee low errors on Hessians (Yuan et al., 2024), but requires dedicated training (Zhao et al., 2025; Cui et al., 2025). Second, compared to energies and forces, Hessians are expensive: Given a probing vector $\mathbf{v}$, AD only lets us calculate $\mathbf{Hv}$. To calculate the full $3N \times 3N$ Hessian, we need $3N$ Hessian vector products (HVPs) with the unit vectors $\mathbf{v} = \mathbf{e}_i, i = 0, ..., 3N - 1$, each yielding one column of the Hessian. Since the energy calculation with an MLIP is usually $O(N)$, the total cost of the Hessian calculation is $O(N^2)$. While the HVPs can theoretically be parallelized, the $O(N^2)$ memory footprint requires even medium-sized systems to be computed sequentially. Rodriguez et al. (2025) in particular showed that including the Hessian in the loss leads to better extrapolation of energy and force predictions, less data for a given accuracy, and improved stability in MD. This comes at the cost of approximately 25 longer training times, which makes AD Hessians prohibitive to train.

## 2.2 OUR APPROACH: DIRECT PREDICTION OF HESSIANS

Instead of relying on automatic differentiation, we propose to directly predict Hessians with an equivariant neural network. Similarly, direct force prediction has been proposed before (Gasteiger et al., 2021; Liao et al., 2024). While direct force prediction can achieve low errors on benchmarks, AD forces are often preferred (Bigi et al., 2025) as they are guaranteed to be conservative. This prevents unphysical energy drift in molecular dynamics simulations, which would produce incorrect observables. We want to stress that the situation is *different* for Hessians. There are no mainstream tasks, that require to integrate Hessians for a long time to obtain a property. Nonetheless, it is important to understand in which situations integrability is important, and when the non-integrability error acts the same as normal approximation errors. We discuss this in more detail in A.2. Another difference between the direct prediction of forces vs Hessians regards the computational speedup achieved: the speed-up of direct force-prediction is "only" about $2\times$-$3\times$ over AD forces, therefore not "worth it" in many cases. In contrast, the savings for direct Hessian predictions can be multiple orders of magnitude, as we will see in section 4.

## 2.3 MOLECULAR OPTIMIZATION WITH HESSIANS

**Rational function optimization (RFO)** Geometry optimization locates minima $\arg \min_{\mathbf{R}} E(\mathbf{R})$ of the potential energy surface (PES) starting from non-equilibrium molecular geometries. It underpins most workflows from global searches to high-throughput screening. If Hessians are available, second-order methods are favored due to their fast convergence guarantees (Nocedal & Wright, 2006; Doikov et al., 2023). RFO (Simons et al., 1983; Banerjee et al., 1985) is a commonly used second-order optimizer (it makes use of Hessian information to accelerate convergence) for molecular geometries (Hermes et al., 2022). An attractive property of RFO is that it can be used both for minimizing $E(\mathbf{R})$ as well as for finding saddle points, and it is robust to indefinite Hessians (Banerjee et al., 1985). As computing the Hessian is usually too expensive, a common practice is to maintain an approximate Hessian using the BFGS quasi-Newton scheme. We provide more details in A.1.3.

**Zero-point energy (ZPE)** Zero-point corrections account for the quantum mechanical vibrational energy that molecules have at absolute zero temperature. To calculate the ZPE, one relaxes a

geometry to an extrema, computes the Hessian, and sums the frequencies $ZPE = \frac{\hbar}{2}\sum_i \sqrt{\tilde{\lambda}_i}$. Here, $\tilde{\lambda}_i$ are eigenvalues of the mass-weighted Hessian $\tilde{\mathbf{H}}_{AB} = \mathbf{H}_{AB}/m_A m_B$, and $\hbar$ is Planck's reduced constant. One is usually interested in the relative ZPE between reactant and product states: $\Delta ZPE = ZPE(\mathbf{R}_R) - ZPE(\mathbf{R}_P)$. The relative ZPE is relevant for reaction thermochemistry, as $\Delta ZPE$ enters as a correction to the reaction free energy $\Delta G$ and therefore the equilibrium constant, which relates forward and backward rates via detailed balance.

**Transition states**   Transition states (TS) are first-order saddle points on the PES. They represent the maximum barrier along the minimum energy path (MEP) between two minima. Identifying transition states is crucial for understanding reaction mechanisms and mapping reaction networks. A first-order saddle point is characterized by having exactly one negative eigenvalue of the Hessian. Over the years, various computational methods have been developed to describe MEPs and locate transition states. We review methods for transition state search in A.1.4.

**Frequency analysis for extrema classification**   To carry out frequency analysis, one must first remove the five or six redundant degrees of freedom, corresponding to the invariance of the energy under rotation and translation. This is done by mass-weighing the Hessian and performing an Eckart projection (Louck & Galbraith, 1976), which we describe in A.6.4. Then, the projected matrix is decomposed into its eigenvalues, with all positive eigenvalues signaling a minimum, and exactly $N$ negative eigenvalue signaling the presence of a order-$N$ transition state.

## 3   HIP HESSIAN PREDICTION

Any equivariant neural network (Duval et al., 2024) with features of at least $l = 2$ can be equipped with a Hessian readout head. In this work, we pick the EquiformerV2 architecture as the backbone with four transformer layers (Liao et al., 2024). Each layer contains a layer norm, message passing with graph attention, another layer norm, and a feed-forward layer.

### 3.1   HESSIAN SYMMETRY REQUIREMENTS

Hessians are symmetric, real-valued, Cartesian tensors. This means, under rotation of the coordinate system with rotation matrix $\mathbf{Q}$, each Hessian sub-block $\mathbf{H}_{I,J}$ transforms as

$$\mathbf{H}_{I,J} \xrightarrow{\mathbf{Q}} \mathbf{Q}\mathbf{H}_{I,J}\mathbf{Q}^\top \tag{5}$$

Any physically meaningful Hessian has to satisfy this rotation symmetry, as well as the symmetry of the upper and lower triangular $\mathbf{H} = \mathbf{H}^\top$. We elaborate on the symmetry of Hessians in A.2.

### 3.2   HESSIAN PREDICTION HEAD

Starting from $T$ layers of message passing in the backbone, we use the atom features $\mathbf{h}_{I,m,c}^{(T)}$, and feed them into a Hessian readout module. Here, we first add another Hessian readout-specific EquiformerV2 layer to improve expressivity. We then construct the Hessian sub-blocks: We first send normal messages with equation 4, but instead of attention and summing, we treat the messages as atom-pair features:

$$\mathbf{h}_{I,J,l,m,c} = \mathbf{v}_{I,J,l,m,c} = \mathbf{v}\big(\mathbf{h}_I^{(T)}, \mathbf{h}_J^{(T)}, \mathbf{r}_{I,J}\big) \tag{6}$$

As we are reusing the message passing machinery with an interaction cutoff radius, predicting Hessians scales initially $O(N)^2$ in memory and compute within the cutoff, and then reduces to $O(N)$ scaling for larger systems. This sparsity with distance assumption aligns with the sparsity of Hessians due to locality of electron interactions that is common in quantum chemistry and can be mathematically rigorously justified (Kussmann et al., 2015). After message passing we have to transform the atom pair features $\mathbf{h}_{I,J,l,m,c}$ into the corresponding Hessian sub-blocks $\mathbf{H}_{I,J}$. First, we project the pair feature irreps down to a single $\{\tilde{\mathbf{h}}_{I,J,l,m}\}_{l\in\{0,1,2\}}$ irreps feature (`1x0e + 1x1e + 1x2e` in `e3nn` notation (Geiger & Smidt, 2022)) using a linear layer $\mathbf{W}_{l,c}$: $\tilde{\mathbf{h}}_{I,J,l,m} = \mathbf{W}_{l,c}\mathbf{h}_{I,J,l,m,c}$ We then expand $\tilde{\mathbf{h}}_{I,J,l,m}$ to an intermediate Hessian sub-block $\mathbf{H}'_{I,J}$ using the tensor product expansion

(Unke et al., 2021):

$$\mathbf{H}'_{I,J,m_1,m_2} = \sum_{l,m} \mathbf{C}^{l,m}_{l_1=1,m_1,l_2=1,m_2} \tilde{\mathbf{h}}_{I,J,l,m} \tag{7}$$

where $\mathbf{C}^{l,m}_{l_1,m_1,l_2,m_2}$ are the Clebsch-Gordon coefficients ensuring that the relation in equation 5 is enforced (Unke et al., 2021). Physically, this is a simple change of basis from the coupled to the uncoupled angular momentum basis (Edmonds, 1996). $\mathbf{H}'_{I,J,m_1,m_2}$ is a $3 \times 3$ block, since $m_1$ and $m_2$ run from $-l_1 = -l_2 = -1$ to $l_1 = l_2 = 1$. As $\mathbf{C}^{l,m}_{l_1=1,m_1,l_2=1,m_2} = 0$ for all $l > 2$, $\mathbf{h}_{I,J,l,m,c}$ only needs to contain irreps up to $l = 2$ to build the $3 \times 3$ Hessian sub-blocks. Intuitively, equation 7 is the inverse of the irreducible tensor decomposition, so it assembles a spherical tensor back from its irreducible components. Conversely, this also means we need all irreps up to $l = 2$ as only including irreps up to $l = 1$ would not be sufficient to express the $3 \times 3$ tensors that decompose into $l = 2$ irreps. Finally, we symmetrize the immediate Hessian to the final prediction with $\mathbf{H} = \mathbf{H}' + \mathbf{H}'^\top$.

## 3.3 LOSS FUNCTION DESIGN

To learn the Hessians, one can use standard loss functions like mean absolute error (MAE) or mean squared error (MSE) between predicted and actual Hessian elements:

$$\mathcal{L}_{\text{MAE / MSE}} = \sum_{i,j} |\mathbf{H}_{i,j} - \mathbf{H}^{\text{pred}}_{i,j}|_{\text{MAE / MSE}} \tag{8}$$

However, we are often mainly interested in the smallest eigenvalues and the corresponding eigenvectors (Besalú & Bofill, 1998; Cerjan & Miller, 1981). For this reason, we use a loss function that emphasize the relevant subspace of Hessian. This loss function is similar to Li et al. (2025), which uses it to emphasize the subspace up to the Lowest Unoccupied Molecular Orbital (LUMO) in Fock matrix prediction. Let $\mathbf{V} = [\mathbf{v}_1, \mathbf{v}_2, ..., \mathbf{v}_{3N}]$ be the matrix with columns made from eigenvectors of the ground truth Hessian $\mathbf{H}$, and $\mathbf{\Lambda}$ the corresponding matrix with eigenvalues on the diagonal and zeros everywhere else. Further denote $\mathbf{V}_{[:,:k]}$ and $\mathbf{V}_{[:,k:]}$ the matrices sliced to contain all columns up to/starting from the $k^{th}$ one. We can then define a subspace loss by

$$\mathcal{L}_{\text{sub}} = \sum_{i,j} \left| \mathbf{V}^\top_{[:,:k]} \mathbf{H}^{\text{pred}} \mathbf{V}_{[:,:k]} - \mathbf{\Lambda}_{[:,:k]} \right|_{i,j} \tag{9}$$

If $\mathbf{H}^{\text{pred}} = \mathbf{H}$ we have $\mathbf{V}^\top \mathbf{H}^{\text{pred}} \mathbf{V} = \mathbf{\Lambda}$, and the loss has therefore the correct minimum. Using $\mathcal{L} = \mathcal{L}_{\text{MAE/MSE}} + \alpha \mathcal{L}_{\text{sub}}$ lets us emphasize the subspace corresponding to the $k$ lowest eigenvectors and eigenvalues. Since the Hessian always has six redundant degrees of freedom with eigenvalues close to zero, we need to set $k > 6$. In practice, we use $k = 8$. As we will show in A.5.1, predicting Hessians with HIP using any loss function significantly outperforms AD Hessians. MAE combined with the subspace loss in equation 9 further improves results in transition state search and extrema classification with frequency analysis by a small margin, which is why we use them in the experiment of the main text. For details and ablations of the choice of loss function see A.5.1.

## 4 EXPERIMENTS

We evaluate our proposed approach for direct prediction of molecular Hessians across multiple tasks that test both accuracy and practical utility. Specifically, we measure prediction accuracy and quantify computational speed-ups relative to AD and finite-difference Hessian calculation. We further assess downstream performance in geometry optimization, zero-point energy, and TS searches. For TS searches, we additionally perform frequency analyses to evaluate the reliability of the predicted Hessians in distinguishing true transition states.

**Dataset** For the main results of the paper we consider the HORM dataset (Cui et al., 2025) of molecular geometries with energy, force, and Hessian labels recomputed at the $\omega$B97X/6-31G* level of theory. The training dataset comprises 1,725,362 molecules from the Transition1x (T1x) dataset with 50,844 molecules from T1x and 60,000 molecules from the RGD1 dataset serving as validation. To isolate the Hessian from the forces and energy, we train only the Hessian prediction head using

Table 1: **Comparison of Hessian prediction errors on the HORM-T1x validation set.** Denoted are the mean absolute errors (MAE), cosine similarity (CosSim), and average time per forward pass on a RTX 3060 GPU. $\lambda_1$ and $v_1$ were obtained after Eckart-projection.

| Hessian | Model | Hessian ↓ eV/Å$^2$ | Eigenvalues ↓ eV/Å$^2$ | CosSim $v_1$ ↑ unitless | $\lambda_1$ ↓ eV/Å$^2$ | Time ↓ ms |
|---|---|---|---|---|---|---|
| AD | AlphaNet | 0.385 | 2.871 | 0.900 | 0.284 | 767.0 |
| | LEFTNet-CF | 0.150 | 0.684 | 0.943 | 0.112 | 1110.7 |
| | LEFTNet-DF | 0.197 | 0.669 | 0.930 | 0.148 | 341.3 |
| | EquiformerV2 | 0.074 | 0.242 | 0.915 | 0.097 | 633.0 |
| | EquiformerV2 (E-F) | 2.252 | 20.752 | 0.279 | 1.372 | 613.3 |
| Predicted | HIP-EquiformerV2 | 0.030 | 0.063 | 0.982 | 0.031 | 38.5 |
| | HIP-EquiformerV2 (end-to-end) | **0.020** | **0.041** | **0.982** | **0.031** | **31.4** |

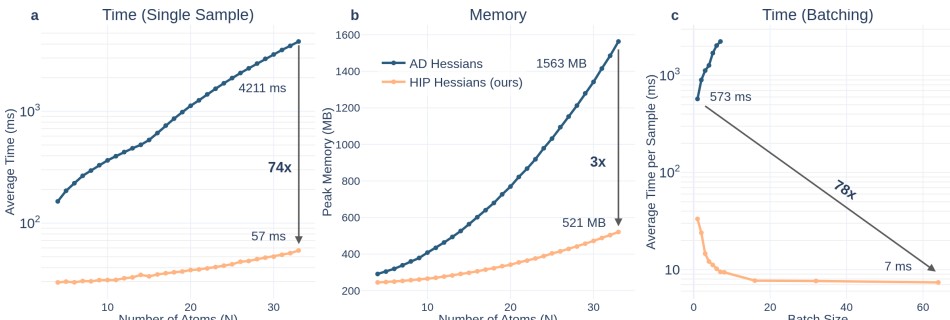

Figure 2: **Computational cost.** (a) Speed and (b) memory footprint for a single sample as a function of molecule size, and (c) scaling of Hessian predictions in parallel with increasing batch size. Timings were performed on an RTX 3060 GPU. AD values are shown in blue, and HIP (this work) in orange.

a fixed EquiformerV2 backbone from HORM (Cui et al., 2025). This ensures that the energy and force predictions of AD-EquiformerV2 and HIP-EquiformerV2 are the same for all experiments. All baseline models originate from Cui et al. (2025) and were also trained using the same ground-truth DFT Hessians. Therefore, all downstream task improvements are purely due to better Hessians.

**Accuracy**    To quantitatively assess the accuracy and speed of our directly predicted Hessians, we calculate the average elementwise mean absolute error, average eigenvalue error, and average time per prediction. Since, for many tasks such as ZPE and TS search, the smallest eigenvalue/eigenvector pair is particularly important, we calculate the cosine similarity and eigenvalue error for these tasks separately. The results are presented in Table 1. Models trained on just energy and forces without Hessian information fail completely (EquiformerV2 (E-F) and Cui et al. (2025)). In every metric, our direct prediction Hessian approach outperforms the other models. The improvement is especially notable in prediction speed, where we are an order of magnitude faster than the models relying on automatic differentiation. Training HIP end-to-end improves the results further.

**Speedup and memory**    In Figure 2(a-b), we further compare the prediction time and memory footprint as a function of molecule size. The direct prediction of Hessians is 10 - 74× faster than using AD for molecules in the HORM dataset. HIP predicted Hessians also benefit from a much more favorable scaling with the molecule size. Figure 2(c) compares the batched AD Hessian as implemented in (Cui et al., 2025) with our HIP-EquiformerV2. We observe at minimum a 78× speedup as a function of batch size compared to AD. We explain the significant performance degradation of batched AD Hessians in A.3.

**Geometry optimization**    We now turn to downstream applications of the Hessian, first focusing on geometry optimization. We compare exact second-order approaches using Hessians (RFO) against first-order methods, and quasi-second-order RFO initialized with Hessians followed by BFGS updates. See A.6.1 for details. Figure 3(a) shows that RFO with HIP Hessians converges within the fewest steps in the median case. In contrast, both finite difference Hessians and AD Hessians frequently fail to converge. Among the Hessian-initialized BFGS methods (blue), the predicted Hessian also performs the best. Steepest descent with line search does not converge within the step budget, which

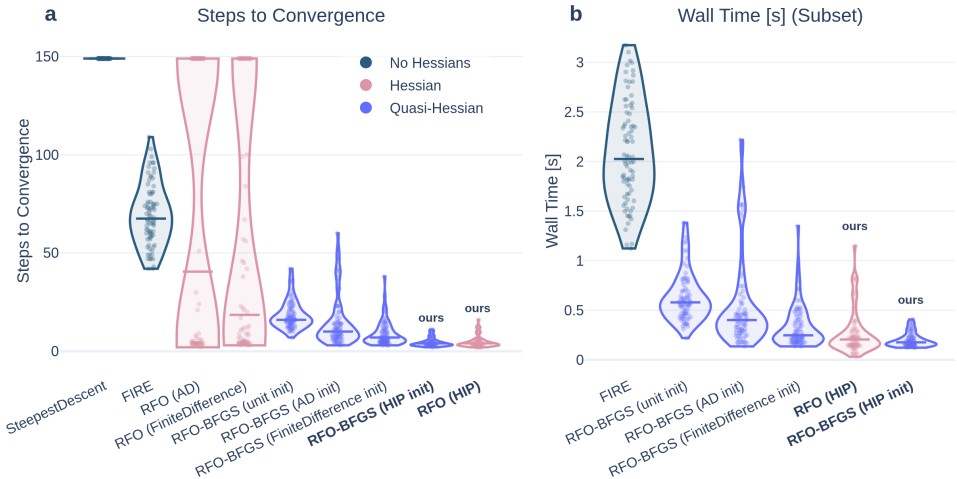

Figure 3: **Geometry optimization.** Methods ordered by their median (horizontal line) for (a) number of steps and (b) wall-clock time to convergence. For clarity, we omit to plot the wall-clock time of RFO with AD or finite difference Hessians, as they require one and two orders of magnitude more time than RFO with HIP predicted Hessians.

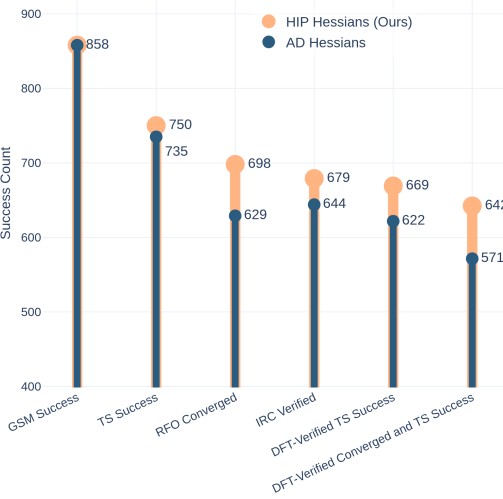

| Hessian | Model | ZPE MAE [eV] | ΔZPE MAE [eV] |
|---------|-------|--------------|---------------|
| AD | AlphaNet | 0.0565 (0.0283) | 0.0625 (0.0770) |
| AD | LeftNet-CF | 1.2957 (0.3331) | 0.1996 (0.2552) |
| AD | LeftNet-DF | 0.0094 (0.0039) | 0.0269 (0.0264) |
| AD | EquiformerV2 | 0.0600 (0.0719) | 0.0539 (0.0706) |
| Predicted | HIP-EquiformerV2 | **0.0004 (0.0003)** | **0.0016 (0.0018)** |

Table 2: **Zero-point energy.** Mean average error and standard deviation of the signed error of the ZPE of the predicted/AD Hessian compared to DFT. We report both the ZPE at the reactant, as well as the relative ΔZPE between reactant and product.

| Hessian | Model | Accuracy ↑ |
|---------|-------|-----------|
| AD | AlphaNet | 71% |
| AD | LeftNet-DF | 78% |
| AD | LeftNet-CF | 82% |
| AD | EquiformerV2 | 75% |
| Predicted | HIP-EquiformerV2 | **92%** |

Figure 4: **Transition state search workflow success rate across diverse models.** "DFT-verified TS success" is defined when the DFT Hessian has one negative eigenvalue. "DFT-verified converged and TS success" additionally requires the DFT force RMS to be below $2 \times 10^{-3}$ Ha/Bohr.

Table 3: **Classifying extrema using Hessian frequency analysis.** We report the accuracy by model in characterizing stationary points via the correct number of negative Hessian eigenvalues. Percentages were computed over 1000 samples, roughly half of which were true transition states and 37% were minima. Validated against DFT.

demonstrates the difficulty of the task. As seen in Figure 3(b), RFO and BFGS methods with HIP Hessians are also the fastest in terms of wall-clock time.

**Zero-point energy** We evaluated an important thermochemical property of the relaxed equilibrium state, the zero-point energy (ZPE). We report both the ZPE of the reactant, as well as the relative ZPE between reactant and product ΔZPE. Experimental details are in A.6.3. Table 2 shows that the HIP-predicted Hessians reach an order of magnitude lower MAE than the next-best model, and two orders of magnitude lower MAE than the same model using automatic differentiation Hessians.

**Transition state search** Another important application of Hessians is transition state search. We use the recently introduced ReactBench benchmark (Zhao et al., 2025), but additionally verify the

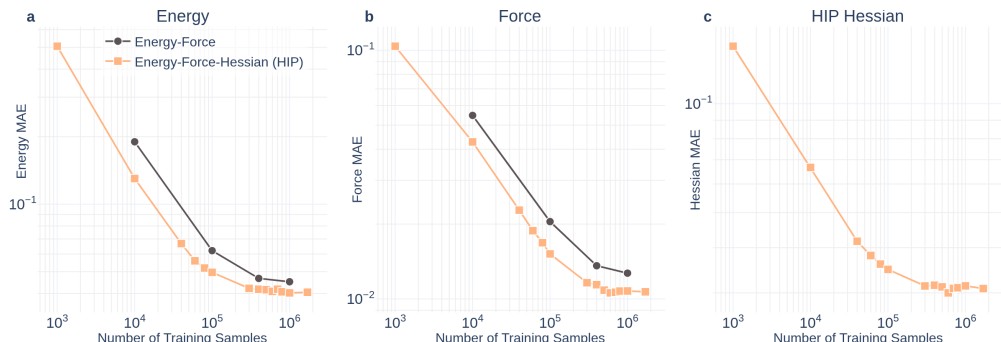

Figure 5: **Accuracy scaling with dataset size.** (a) Energy (b) Force (c) Hessian MAE on the HORM-T1x validation over the number of training samples randomly chosen from the HORM-T1x train set. EquiformerV2 trained on energy and forces is shown in dark gray, the same model trained as HIP also on Hessian data (this work) in orange.

predicted transition states with DFT. A step-by-step description of the workflow can be found in A.6.2. We present metrics of each step of the workflow in Figure 4. The growing string method (GSM) success rate is the same for both AD and HIP-EquiformerV2 models. GSM only uses energy and forces, which are the same for both models. For all subsequent metrics, the HIP predicted Hessians outperform the AD ones. The RFO convergence and DFT-verified convergence metrics depend on the Hessian the most. For both, the HIP Hessians yield the most improvements over AD Hessians.

**Frequency analysis for extrema classification** We further investigate the ability of the Hessian methods to differentiate different extrema on the potential energy surface. We perform frequency analysis on 1000 geometries from the HORM-T1x validation set. Roughly half (44%) are 1st order transition states, 37% are minima, and the rest are higher order transition states according to their DFT Hessians. As shown in Table 2, the HIP-predicted Hessians achieve the best accuracy compared to all models using AD. In particular, HIP-EquiformerV2 has a significant advantage of 10 percentage points in accuracy over any other model, and +17% over the same model with AD (EquiformerV2).

**End-to-end HIP training improves energies and forces** So far we restricted ourselves to only finetuning the Hessian prediction head while freezing the existing parameters. We now investigate the more realistic decision of training either a regular MLIP on energy and forces, or a HIP including the Hessian prediction. We train both model types for the same wallclock time of 3 days at various data scales. Figure 5 shows that end-to-end training of the joint backbone and the readout not only improves the Hessian, but also the energy and force predictions.

**Limitations** Direct Hessian regression requires ground-truth data from DFT, similar to the energy and force labels used for regular MLIPs. The experiments in this work focus on small organic molecules, and future work should explore applying Hessian prediction to other chemical systems, such as materials, larger complexes, or even proteins. To isolate the Hessian from the force and energy predictions, we trained only the Hessian regression head using a fixed backbone. For optimal results on all benchmarks, energy, forces, and Hessians should be trained end-to-end.

## 5 CONCLUSION

In this work, we have presented Hessian Interatomic Potentials (HIP), a novel method for directly predicting molecular Hessians using SE(3)-equivariant neural networks. Our approach eliminates the need for traditional computationally expensive methods, such as finite differences or automatic differentiation, offering a significant improvement in computational time, memory usage, and accuracy. Through extensive validation across a range of critical molecular tasks, we have shown that our predicted Hessians are highly effective for practical applications in computational chemistry. By making Hessians more accessible, we believe our method enables new possibilities for high-throughput screening, material discovery, and drug design. For future work, any base model using spherical harmonics and message passing can be made HIP.

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

# A APPENDIX

## A.1 EXTENDED BACKGROUND

### A.1.1 ANALYTICAL HESSIANS VIA CPKS

The gold standard for computing nuclear Hessians is to derive them analytically. In Kohn-Sham DFT, this is done with the Coupled Perturbation Kohn-Sham (CPKS) method. This is much more complex and computationally demanding than energy or force calculations. The difficulty arises from the need to calculate the response term of the electron density, which involves understanding how the self-consistent field (SCF) solution to the Kohn-Sham equations changes with a change in the nuclear coordinates. To see why this is necessary, write the DFT energy in terms of the density matrix $\mathbf{P}(\mathbf{R}) = (\mathbf{C})^T(\mathbf{R})\mathbf{S}(\mathbf{R})\mathbf{C}(\mathbf{R})$ with molecular orbital coefficients $\mathbf{C}$ and overlap matrix $\mathbf{S}$:

$$E[\mathbf{P}] = \text{Tr}[h\mathbf{P}] + \frac{1}{2}\text{Tr}[\mathbf{J}[\mathbf{P}]] + \rho(\mathbf{P}) \tag{10}$$

To get the force, so the nuclear gradient, we first have to define the Lagrangian to enforce normalization of the wavefunction:

$$\mathcal{L}(C, \epsilon, \mathbf{R}) = E[\mathbf{R}] - \text{Tr}[\epsilon(\mathbf{C}^T\mathbf{S}\mathbf{C} - \mathbf{I})] \tag{11}$$

We then get the nuclear gradient by differentiating $\mathcal{L}$. The derivative w.r.t. nuclear component $x$ is:

$$\frac{d\mathcal{L}}{dx} = \frac{\partial\mathcal{L}}{\partial x} + \underbrace{\frac{\partial\mathcal{L}}{\partial \mathbf{C}}}_{=0}\frac{\partial \mathbf{C}}{\partial x} - \text{Tr}\left[\epsilon\mathbf{C}^T\frac{\partial\mathbf{S}}{\partial x}\mathbf{C}\right] + 2\text{Tr}\left[(\mathbf{S}\mathbf{C}\epsilon)^T\frac{d\mathbf{C}}{dx}\right] \tag{12}$$

$$= \frac{\partial E}{\partial x} - 2\text{Tr}\left[(\mathbf{S}\mathbf{C}\epsilon)^T\frac{dC}{dx}\right] - \text{Tr}\left[\epsilon\mathbf{C}^T\frac{\partial\mathbf{S}}{\partial x}\mathbf{C}\right] + 2\text{Tr}\left[(\mathbf{S}\mathbf{C}\epsilon)^T\frac{dC}{dx}\right] \tag{13}$$

$$= \frac{\partial E}{\partial x} - \underbrace{\text{Tr}\left[\epsilon\mathbf{C}^T\frac{\partial\mathbf{S}}{\partial x}\mathbf{C}\right]}_{\text{"Pulay terms"}} \tag{14}$$

Due to the stationarity $\frac{\partial\mathcal{L}}{\partial C} = 0$ we did not need to calculate the density response $\frac{\partial C}{\partial x}$, which is why calculating forces is cheap, on the same order as the cost of energies. However, once we want Hessians, we need to differentiate equation 11 twice:

$$\frac{d^2\mathcal{L}}{dxdy} = \frac{\partial^2\mathcal{L}}{\partial x\partial y} + \frac{\partial^2\mathcal{L}}{\partial\mathbf{C}\partial x}\frac{d\mathbf{C}}{dy} + \frac{\partial^2\mathcal{L}}{\partial\epsilon\partial x}\frac{d\epsilon}{dy} \tag{15}$$

$$= \frac{\partial^2\mathcal{L}}{\partial y\,\partial x} - \begin{bmatrix}\frac{\partial^2\mathcal{L}}{\partial\mathbf{C}\,\partial x} & \frac{\partial^2\mathcal{L}}{\partial\varepsilon\,\partial x}\end{bmatrix}\begin{bmatrix}\frac{\partial^2\mathcal{L}}{\partial\mathbf{C}^2} & \frac{\partial^2\mathcal{L}}{\partial\mathbf{C}\,\partial\varepsilon} \\ \frac{\partial^2\mathcal{L}}{\partial\varepsilon\,\partial\mathbf{C}} & \frac{\partial^2\mathcal{L}}{\partial\varepsilon^2}\end{bmatrix}^{-1}\begin{bmatrix}\frac{\partial^2\mathcal{L}}{\partial\mathbf{C}\,\partial y} \\ \frac{\partial^2\mathcal{L}}{\partial\varepsilon\,\partial y}\end{bmatrix}. \tag{16}$$

Solving the linear system of equations scales formally as $O(N^5)$ in computation and $O(N^4)$ in memory. Similar to energy calculations, density fitting, sparsification (integral screening), and other optimizations can reduce the cost for CPKS. Still, the scaling remains worse in both computation and memory compared to energy calculations.

### A.1.2 HESSIANS VIA FINITE DIFFERENCES

A major downside of the CPKS approach to Hessians is its highly complex implementation and need for exchange-functional-specific derivations. Therefore, many new methods do not support Hessians, and most codes instead resort to finite difference calculations in these cases. Using central differences of the forces along the $3N$ Cartesian directions, the Hessian elements are constructed from

$$\mathbf{H}_{ij} \approx \frac{\mathbf{F}_i(\mathbf{R} + h\mathbf{e}_j) - \mathbf{F}_i(\mathbf{R} - h\mathbf{e}_j)}{2h} \tag{17}$$

This requires $2 \times 3N$ displaced *gradient* calculations. Each gradient calculation scales similarly to the energy calculation with $O(N^4)$, leading to a $O(N^5)$, the same as naive CPKS, although typically with more numerical noise and a much larger prefactor because we have to repeatedly converge an SCF, as opposed to solving a large linear system in analytical Hessians.

### A.1.3 GEOMETRY OPTIMIZATION

Geometry optimization locates minima $\arg\min_{\mathbf{R}} E(\mathbf{R})$ of the potential energy surface starting from non-equilibrium molecular geometries. It underpins most workflows from global searches to high-throughput screening. The efficiency of geometry optimization depends on four key factors: (i) the choice of coordinate system, (ii) the quality of the Hessian approximation, (iii) the update strategy, and (iv) the step size control. Cartesian coordinates are straightforward but strongly coupled, which limits step size and slows convergence for flexible systems. In contrast, internal coordinates are better aligned with chemically meaningful motions, separate stiff and soft modes, and allow constraints to be imposed naturally. These properties often lead to substantially fewer iterations. Some of the most widely used optimizers are the first-order FIRE (Bitzek et al., 2006) optimizer and the second-order rational function optimization (RFO) optimizer.

**RFO**   Rational function optimization (RFO) (Simons et al., 1983; Banerjee et al., 1985) is a commonly used second-order optimization technique in molecular geometry optimization. RFO starts with a [2/2]-Padé-expansion of the energy:

$$E(\mathbf{R_t} + \Delta\mathbf{x}) - E(\mathbf{R_t}) \approx \frac{\mathbf{g}^\top \Delta\mathbf{x} + \frac{1}{2}\Delta\mathbf{x}^\top \mathbf{H}\Delta\mathbf{x}}{1 + |\Delta\mathbf{x}|^2} \tag{18}$$

The extrema $\Delta x'$ of this surrogate are given by the solution of the generalized eigenvalue problem:

$$\begin{bmatrix} \mathbf{H} & \mathbf{g} \\ \mathbf{g}^\top & 0 \end{bmatrix} \begin{bmatrix} \Delta\mathbf{x}' \\ 1 \end{bmatrix} = \lambda \begin{bmatrix} I & 0 \\ 0 & 1 \end{bmatrix} \begin{bmatrix} \Delta\mathbf{x}' \\ 1 \end{bmatrix}. \tag{19}$$

An attractive property of RFO is that it can be used both for minimizing $E(\mathbf{R})$ as well as for finding saddle points (important for transition state search, see next section): If we pick the first eigenvector of equation 19, we get an update pointing to the minimum; if we select the second eigenvector, we get a transition state update. In contrast to Newton-Raphson optimization, RFO is also robust to indefinite Hessians. For detailed derivation, see (Simons et al., 1983; Banerjee et al., 1985). In this paper, for minimization, we are using restricted step RFO (RS-RFO), which adds a trust region to prevent unphysically large steps. For the transition point search, we are using restricted step partitioned RFO (RS-P-RFO), a slight modification that treats the subspace with negative and positive eigenvalues of the Hessian separately.

**BFGS**   To apply RFO in practice, we need a Hessian at each step. As computing full Hessians is usually too expensive, a common practice is to maintain an approximate Hessian using the BFGS quasi-Newton scheme. BFGS updates approximate Hessians $\mathbf{B}_k$ of the true Hessian $\mathbf{H}(\mathbf{R}_k)$ using the update equation

$$\mathbf{B}_{k+1} = \mathbf{B}_k - \frac{\mathbf{B}_k \mathbf{s}_k \mathbf{s}_k^\top \mathbf{B}_k}{\mathbf{s}_k^\top \mathbf{B}_k \mathbf{s}_k} + \frac{\mathbf{y}_k \mathbf{y}_k^\top}{\mathbf{y}_k^\top \mathbf{s}_k}, \qquad \mathbf{y}_k^\top \mathbf{s}_k > 0, \tag{20}$$

where the curvature condition $\mathbf{y}_k^\top \mathbf{s}_k > 0$ (typically ensured by a Wolfe line search) guarantees that $\mathbf{B}_{k+1}$ remains positive-definite, which is desirable for minimization. In the RFO framework, $\mathbf{B}_k$ simply replaces the exact Hessian $\mathbf{H}$ in the augmented eigenvalue problem in equation 19.

### A.1.4 TRANSITION STATE SEARCH

Computational methods for transition state search can broadly be categorized as *single-ended* and *double-ended*. In double-ended methods, we know the product and reactant states (the two minima on the energy surface) and try to find an interpolation on the MEP that goes through the TS. In contrast, in single-ended methods, we only know a starting point and try to climb up on the energy surface to find a nearby transition state. Often, double-ended methods are used to give good initial guesses, which are then refined by single-ended methods. In this study, we are using the Growing string method to find the initial states, which we then refine with RS-P-RFO (see above).
Different double-ended methods are distinguished by how they grow the interpolations between product and reactant states. The most common approaches are the nudge elastic band method (Henkelman & Jónsson, 1999; Henkelman et al., 2000) and the growing string method (GSM) (Peters et al., 2004; Zimmerman, 2015).

**Nudged elastic band method**  The nudged elastic band method aims to find the minimum energy path between given initial and final configurations. A set of images connecting these states is linked by classical spring forces, forming an elastic band. Each image experiences a total force comprising the spring force along the local tangent and the true force perpendicular to it. The images are simultaneously optimized to trace the MEP. To converge to the transition state, a climbing image is introduced, for which the force along the band is inverted, leaving only the perpendicular component of the true force. This drives the climbing image up the potential energy surface along the band while descending perpendicularly, reaching the transition state along the MEP.

**Growing string method**  The growing string method (GSM) locates transition states by incrementally constructing a discrete path of structures on the potential energy surface. Starting from the reactant and product endpoints, two path fragments are iteratively grown together. At each iteration, a new node on the string is added along the local tangent direction and relaxed orthogonally to the path according to the force

$$\mathbf{F}_\perp = \mathbf{F} - (\hat{\mathbf{t}}^\top \mathbf{F})\hat{\mathbf{t}}, \tag{21}$$

where $\mathbf{F} = -\nabla E$ is the force and $\hat{\mathbf{t}}$ is the local tangent unit vector. Convergence is typically accelerated by approximate Hessians, constructed in delocalized internal coordinates and updated by quasi-Newton schemes, which enable eigenvector-guided optimization while avoiding full second-derivative evaluations (Zimmerman, 2015). Once the two fragments merge, the full string is reparameterized to maintain uniform node distance, and the highest energy node serves as a transition state estimate, which can be further refined using Hessian-based eigenvector following (Peters et al., 2004).

A.2  HESSIAN PROPERTIES

The Hessian is a symmetric matrix $\mathbf{H} = \mathbf{H}^\top$ where each sub-block transforms under rotation as a cartesian tensor

$$\mathbf{H}_{I,J} \xrightarrow{\mathbf{Q}} \mathbf{Q}\mathbf{H}_{I,J}\mathbf{Q}^\top \tag{22}$$

or equivalently

$$\mathbf{H} \xrightarrow{\mathbf{Q}} (\mathbf{I}_N \otimes \mathbf{Q})\mathbf{H}(\mathbf{I}_N \otimes \mathbf{Q})^\top \tag{23}$$

The symmetry follows directly from Schwartz's theorem, which states that for every scalar function $E(\mathbf{R})$ that has continuous partial second derivatives in the neighborhood of a point $\mathbf{R_0}$, the partial second derivatives commute

$$\left(\frac{\partial^2 E}{\partial \mathbf{R}_i \partial \mathbf{R}_j}\right)_{\mathbf{R}=\mathbf{R_0}} = \left(\frac{\partial^2 E}{\partial \mathbf{R}_j \partial \mathbf{R}_i}\right)_{\mathbf{R}=\mathbf{R_0}} \tag{24}$$

The transformation law under rotation follows straightforwardly from the chain rule:
First, write the rotated coordinates as

$$\mathbf{R}' = (\mathbf{I}_N \otimes \mathbf{Q})\mathbf{R} \tag{25}$$

$$\frac{\partial \mathbf{R}'}{\partial \mathbf{R}} = (\mathbf{I}_N \otimes \mathbf{Q}) \tag{26}$$

Then, define the energy in the rotated frame

$$E'(\mathbf{R}') = E((\mathbf{I}_N \otimes \mathbf{Q})^\top \mathbf{R}') = E(\mathbf{R}) \tag{27}$$

Then the first order derivative is

$$\nabla_{\mathbf{R}'} E'(\mathbf{R}') = (\mathbf{I}_N \otimes \mathbf{Q})\nabla_{\mathbf{R}} E'(\mathbf{R}') \tag{28}$$

$$= (\mathbf{I}_N \otimes \mathbf{Q})\nabla_{\mathbf{R}} E((\mathbf{I}_N \otimes \mathbf{Q})^{-1}(\mathbf{I}_N \otimes \mathbf{Q})\mathbf{R}') \tag{29}$$

$$= (\mathbf{I}_N \otimes \mathbf{Q})\nabla_{\mathbf{R}} E(\mathbf{R}) \tag{30}$$

showing the equivariance of the forces. The second-order derivative is

$$\mathbf{H}' = \nabla^2_{\mathbf{R}'} E'(\mathbf{R}') = \nabla_{\mathbf{R}'}((\mathbf{I}_N \otimes \mathbf{Q})\nabla_{\mathbf{R}} E(\mathbf{R})) \tag{31}$$

$$= (\mathbf{I}_N \otimes \mathbf{Q})\nabla_{\mathbf{R}'}\nabla_{\mathbf{R}} E(\mathbf{R}) \tag{32}$$

$$= (\mathbf{I}_N \otimes \mathbf{Q})\nabla_{\mathbf{R}}\nabla_{\mathbf{R}} E(\mathbf{R})(\mathbf{I}_N \otimes \mathbf{Q})^\top \tag{33}$$

$$= (\mathbf{I}_N \otimes \mathbf{Q})\mathbf{H}(\mathbf{I}_N \otimes \mathbf{Q})^\top \tag{34}$$

which shows the equivariance of the Hessian matrix.

**Non-integrability Hessians** Since direct Hessian predicrtions are not formulated as derivatives, they are not guaranteed to be integrable. It is important to understand when a non-integrability error can lead to catastrophically wrong downstream predictions similar to direct force predictions in MD simulations.

Lets first review the more familiar case: Why do direct force predictions fail in MD simulations but succeed in geometry optimization? Suppose we directly predict forces with error $e(x) = F_{\text{pred}}(x) - F(x)$. We can decompose the error in a conservative and a non-conservative field using the Helmholtz decomposition: $e(x) = \nabla\phi(x) + S(x)$. $\phi \in \mathbb{R}$ is a scalar potential, and S is a vector field $S(x) \in \mathbb{R}^{3N}$ that is divergence-free $\nabla \cdot S(x) = 0$ but not curl-free $\exists x$ s.t. $\nabla \times S(x) \neq 0$. Now consider how the energy of the system changes over time in an MD simulation:

$$\Delta E' = \int_0^T F_{\text{pred}}(x(t))v(t)dt = \int_0^T (F(x(t) + e(x(t)))v(t)dt$$

$$= \int_0^T (F(x) + \nabla\phi(x) + S(x))v(t)dt$$

$$= E(T) - E(0) + \underbrace{\phi(T) - \phi(0)}_{\text{error bounded with time}} + \underbrace{\int_0^T S(x)v(t)dt}_{\text{error unbounded with time}}$$

Since MD simulations use large $T$, non-conservative forces lead to the blowups in error.

In geometry optimization we define $E' = E + \phi$, then follow the gradient descent ODE:

$$\frac{dE'(x(t))}{dt} = \nabla E'(x(t)) \cdot \frac{dx(t)}{dt} \tag{35}$$

$$= -\nabla E'(x(t)) \cdot (\nabla E(x) + e(x)) \tag{36}$$

$$= -\nabla E'(x(t)) \cdot (\nabla E(x) + \nabla\phi(x) + S(x)) \tag{37}$$

$$= -\nabla E'(x(t)) \cdot (\nabla E'(x) + S(x)) \tag{38}$$

$$= -|\nabla E'(x(t))|^2 + \nabla E'(x) \cdot S(x) \tag{39}$$

Using the general inequality $2ab \leq a^2 + b^2$:

$$\frac{dE'(x(t))}{dt} = -|\nabla E'(x(t))|^2 + \nabla E'(x) \cdot S(x) \tag{40}$$

$$\leq -|\nabla E'(x(t))|^2 + \frac{1}{2}|\nabla E'(x(t))|^2 + \frac{1}{2}|S(x)|^2 \tag{41}$$

$$= -\frac{1}{2}|\nabla E'(x(t))|^2 + \frac{1}{2}|S(x)|^2 \tag{42}$$

Therefore, as long as $|\nabla E'(x(t))|^2 > |S(x)|^2$, the direct predicted force minimizes the energy plus some error potential similar to AD forces. Once we are close to convergence we have $|\nabla E'(x(t))|^2 \leq |S(x)|^2$ and stop converging or enter a limit cycle. This means, we will converge to an extrema on a energy surface $E'$ within a distance of $O(|S|)$. In practice we are interested in how close we converge to the extrema of the DFT energy $E$. For this it doesnt matter if the error stems from the error potential $\phi$ shifting the extrema, or $S$ inducing a small limit cycle. As there is no time dependent error blow up it is therefore safe to use direct forces for geometry optimization.

Now we can do the same for the directly predicted Hessian $H_{\text{pred}}(x) = H(x) + e(x)$: Denote the error again as $e(x) = \nabla^2\phi(x) + S(x)$. As before we would get errors scaling with integration length if we would try to calculate quantities of the form

$$Q(T) = \int_0^T f(x(t), H_{\text{pred}}(x(t)))dx(t) \tag{43}$$

leading to large errors for large $T$.

However, our key argument is: No mainstream task integrates over a function of the Hessian for large $T$!

ZPE and geometry classification are single point calculation, therefore the errors $\nabla^2\phi(x)$ and $S(x)$ both behave the same, one error is not "worse" than the other.

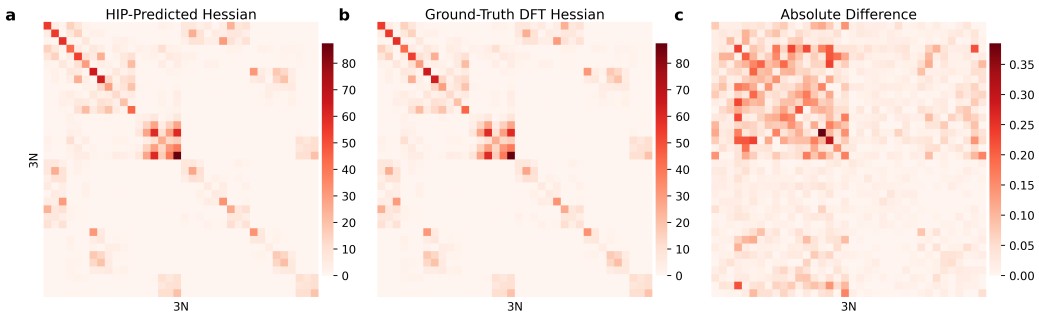

Figure 6: **HIP compared to ground-truth DFT Hessian.** The horizontal and vertical axis index the $(3N_{Atoms})^2$ entries of the Hessian. We depict the absolute values of test sample C5H6O in eV/Å$^2$.

Mode following in transition state search is equivalent to the geometry optimization analysis with direct forces above. Energy minimization with Hessian preconditioner is also unproblematic, as preconditioner cannot change the minimum, they only change the convergence speed.

In fact, it is common practice to use BFGS updates for transition state search and energy minimization, and BFGS updates do not have any notion of integrability either.

### A.3 BATCHED AD HESSIAN PERFORMANCE

We observe in Figure 2 a fast degradation of the AD Hessian speed with batch size. To explain this, we have to understand exactly how AD treats Hessians. From the point of view of the AD engine, there is no difference between a batch of two molecules of system size $N_A$ and $N_B$, and a larger system of size $N_A + N_B$. In both cases, the MLIP function takes in an array of dimension $3N_A + 3N_B$, and returns either a single scalar $E_A + E_B$ or even worse, an array $[E_A, E_B]$, so mathematically the energy function looks like $E(\mathbf{R}) : \mathbb{R}^{3N_A+3N_B} \to \mathbb{R}$ or $E(\mathbf{R}) : \mathbb{R}^{3N_A+3N_B} \to \mathbb{R}^2$. Consequently, a Hessian of this function is a block diagonal matrix of dimension $3(N_A + N_B) \times 3(N_A + N_B)$, or even $2 \times 3(N_A + N_B) \times 3(N_A + N_B)$. Therefore, the memory costs grow quadratically with the batch size. Since we have to implement the Hessian calculation with sequential Hessian-Vector products in order not to run out of memory, we can not even make use of parallelization in the batched processing of Hessians.

### A.4 TRAINING DETAILS

The hyperparameters our HIP-EquiformerV2 model are listed in 4. We inherit the model settings and large parts of the optimizer settings from the HORM codebase (Cui et al., 2025).

**Training resources**  Training AD Hessians is exceptionally more expensive than training HIP. The memory cost of computing loss gradients through the full $3N \times 3N$ Hessian is prohibitive. Instead, the baseline models from Cui et al. (2025) randomly sample one or two columns of the Hessian via random Hessian-vector products at each training step. Despite this, training AD Hessians remains exceedingly expensive. The HORM paper states that they use 2 HVPs at a batch size of 128. Their checkpoint metadata shows that the per device batch size was 8, implying a 16xA30 GPU setup for 1 million gradient steps. In contrast, we trained HIP on a single H100 GPU for 500k steps. Additionally, each train step of HIP costs 1/5 the cost of AD Equiformer training on our hardware. Finally, we observe HIP fully converges after 300-500k training steps, while AD Hessians require a multiple of steps, due to the limited supervision of not being able to use the full Hessian.

### A.5 ADDITIONAL RESULTS

#### A.5.1 LOSS FUNCTION

We are trying to design a loss function that emphasizes the lowest lying eigenvalues/eigenvectors. To design such a loss function, one could naively compute a loss directly as $L = \sum_i^k |\mathbf{v}_i^{\text{predict}} - \mathbf{v}_1^{\text{true}}|$.

Table 4: **Hyperparameters**. We use the same hyperparameters for HIP as HORM (Cui et al., 2025) for simplicity.

| | |
|---|---|
| Model | Type: EquiformerV2 |
| | Layers: 4 |
| | Sphere channels: 128 |
| | Attention hidden channels: 64 |
| | Attention heads: 4 |
| | Attention alpha channels: 64 |
| | Attention value channels: 64 |
| | FFN hidden channels: 128 |
| | Activation: SiLU |
| | Distance basis: 512 (Gaussian) |
| | Radius cutoff: 12 Angstrom |
| | Cutoff Hessian: 12 Angstrom |
| | Hessian layers: 3 (1 for end-to-end) |
| | Spherical harmonics: Lmax=4, Mmax=2 |
| | Grid resolution: 18, Sphere samples: 128 |
| | Dropout: $\alpha$=0, drop path=0 |
| Loss | MAE weight: 1 |
| | Eigen subspace $k$: 8 |
| | Eigen loss weight $\alpha$: 1.0 |
| Optimizer | AdamW |
| | Betas: (0.9, 0.999) |
| | AMSGrad: True |
| | Weight decay: 0 |
| | Batch size: 128 |
| | Gradient clipping: 0.1 (norm) |
| Learning Rate | Learning rate: 0.0005 |
| | Scheduler: StepLR |
| | Step size: 10 |
| | Gamma: 0.85 |
| Trainer | Steps: 500k (3-5 days on a H100 GPU) |

| Model | Hessian Columns | Batch Size | BZ / GPU | GPU | # GPUs | Training Steps |
|---|---|---|---|---|---|---|
| AlphaNet | 1 | 32 | 16 | H20 | 2 | 688,000 |
| LeftNet-CF | 1 | 64 | 16 | A30 | 8 | 192,000 |
| LeftNet-DF | 2 | 64 | 16 | A30 | 8 | 120,000 |
| EquiformerV2 (E-F) | 0 | 128 | 8 | A30 | 16 | 384,000 |
| EquiformerV2 | 2 | 128 | 8 | A30 | 16 | 1,008,000 |
| HIP-EquiformerV2 | 3N (all) | 128 | 128 | H100 | 16 | 500,000 |

Table 5: **Summary of training resources.** Comparing HIP to checkpoints from HORM (Cui et al., 2025).

This requires computing the eigenvalues and eigenvectors of the true and predicted Hessian via an eigenspectrum decomposition. Unfortunately, backpropagation through such an eigenspectrum decomposition is numerically unstable (van der Aa et al., 2007). First, gradients computed using the eigenvectors tensor will only be finite when **H** has distinct eigenvalues. Furthermore, if the distance between any two eigenvalues is close to zero, the gradient will be highly sensitive, as it depends on the eigenvalues $\lambda_i$ through the computation of $\min_{i \neq j} \frac{1}{\lambda_i - \lambda_j}$. Instead, we propose the following subspace loss.

In the following section, we compare the eigenspace loss that we introduced in equation 9 to the standard choice of mean square error (MSE) and mean absolute error (MAE).

In particular, we emphasize the subspace $k = 8$. The reason for this is that transition state search, ZPE, and frequency analysis for extrema classification all depend on the two lowest-lying eigenvectors and eigenvalues. We choose $k = 8$ instead of $k = 2$ to account for the usual 6 redundant degrees of freedom, which we do not remove during training for numerical stability.

| Loss | Hessian ↓ eV/Å$^2$ | Eigenvalues ↓ eV/Å$^2$ | CosSim $v_1$ ↑ unitless | $\lambda_1$ ↓ eV/Å$^2$ | $\lambda_2$ ↓ eV/Å$^2$ | $\lambda_1^E$ ↓ eV/Å$^2$ | $\lambda_2^E$ ↓ eV/Å$^2$ | CosSim $v_1^E$ ↑ unitless | CosSim $v_2^E$ ↑ unitless |
|---|---|---|---|---|---|---|---|---|---|
| MSE | 0.033 | 0.072 | 0.728 | 0.168 | 0.088 | 0.037 | 0.013 | 0.959 | 0.928 |
| MAE | **0.026** | **0.057** | 0.825 | **0.127** | 0.052 | **0.031** | 0.012 | 0.976 | 0.952 |
| MAE+Sub | 0.030 | 0.063 | **0.870** | 0.130 | **0.030** | 0.031 | **0.010** | **0.980** | **0.957** |

Table 6: Accuracy of HIP-EquiformerV2 using different loss functions. Superscript $E$ denotes after removing redundant degrees of freedom using Eckart-projection on the mass-weighted matrix.

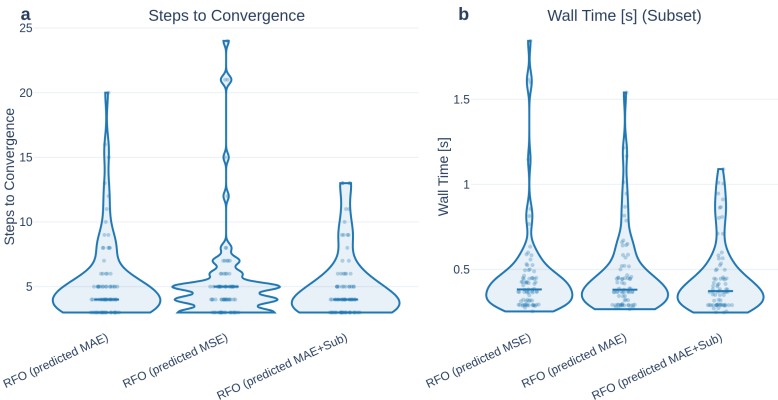

Figure 7: Geometry relaxation with HIP-EquiformerV2 trained using different loss functions

We combine the subspace loss with MAE, so the total loss becomes

$$\mathcal{L}_{\text{MAE+Sub}} = \mathcal{L}_{\text{MAE}} + \mathcal{L}_{\text{sub}}^{(k=8)} \tag{44}$$

$$= \sum_{i,j} |\mathbf{H}_{i,j} - \mathbf{H}_{i,j}^{\text{pred}}| + \sum_{i,j} \left| \mathbf{V}_{[:,:k]}^\top \mathbf{H}^{\text{pred}} \mathbf{V}_{[:,:8]} - \mathbf{\Lambda}_{[:,:8]} \right|_{i,j} \tag{45}$$

For simplicity, we weight both the MAE and subspace loss equally. Future work could benefit from carefully tuning the relative weights between the loss terms.

On downstream tasks, subspace loss improves over MAE and MSE in (i) transition state search, as shown in Table 8, and (ii) transition state identification frequency analysis, as shown in Table 9. We observe equal performance between MAE and the subspace loss for (i) geometry relaxation Table 7 and (ii) computing the zero-point energy Table 7.

Table 6 shows that the subspace loss improves the first eigenvector $v_1$ cosine similarity, and the second eigenvalue $\lambda_2$ MAE. This is consistent with the better performance of the subspace loss on transition state-related tasks, since the transition state is characterized via the two smallest eigenvalues $\lambda_1, \lambda_2$, and is found by following the first eigenvector $v_1$.

MSE generally underperforms compared to MAE and MAE with subspace loss. MSE has the theoretical advantage that it is rotation invariant, while the MAE is not. In practice, however, we find that the MSE leads to a high variance (spikes) in the training loss and gradient norm across batches. We speculate that MAE outperforms MSE due to training stability.

Hessian prediction with any loss function tested significantly outperforms prior AD Hessians.

**RGD1**  To test the extrapolation of HIP we test the same model checkpoints on the slightly larger molecules in the RGD1 dataset in table 10. Similar to T1x, models trained without Hessian data fail

| Hessian | Model | ZPE MAE (Std) [eV] | ΔZPE MAE (Std) [eV] |
|---|---|---|---|
| MSE | HIP-EquiformerV2 | 0.0008 (0.0012) | 0.0019 (0.0025) |
| MAE | HIP-EquiformerV2 | 0.0004 (0.0004) | **0.0013 (0.0019)** |
| MAE+Sub | HIP-EquiformerV2 | **0.0004 (0.0003)** | 0.0016 (0.0018) |

Table 7: Zero-point energy by HIP-EquiformerV2 for different loss functions

| Loss | Model | TS Success | RFO Converged | Both |
|------|-------|-----------|---------------|------|
| MSE | HIP-EquiformerV2 | 718 | 669 | 659 |
| MAE | HIP-EquiformerV2 | 740 | 693 | 683 |
| MAE+Sub | HIP-EquiformerV2 | **750** | **705** | **698** |

Table 8: Transition state search with HIP-EquiformerV2 trained with different loss functions

| Name | TPR ↑ | FPR ↓ | FNR ↓ | TNR ↑ | Precision ↑ | Accuracy ↑ | Accuracy All ↑ |
|------|-------|-------|-------|-------|-------------|------------|----------------|
| MSE | 88% | 11% | 12% | 89% | 86% | 89% | 86% |
| MAE | 93% | 7% | 7% | 93% | 91% | 93% | 91% |
| MAE+Sub | **93%** | **6%** | **7%** | **94%** | **92%** | **94%** | **92%** |

Table 9: Identifying transition states (left) and any extrema (rightmost column) via frequency analysis using HIP-EquiformerV2 for different loss functions.

to produce accurate AD Hessians (EquiformerV2 (E-F)). HIP produces significantly better Hessians faster on every metric compared to AD. End-to-end training HIP improves over only training the readout head. We reach the same conclusions on the QM9 Hessian dataset in A.5.1. Training HIP from scratch is also orders of magnitude cheaper to train than AD Hessians, as we discuss in A.4.

**Computational cost compared to finite difference**   Wander et al. (2025) showed that finite difference Hessians obtained from finetuned MLIPs can increase success rates in transition state search for catalysis. The advantage of finite difference is that it does not require additional memory over a regular forward pass, and only force labels are needed for training. The limitation of finite difference Hessians is their steep time cost. Like AD Hessians, finite difference requires $O(N^2)$ time, but with an even larger prefactor, as can be seen in figure 8.

**QM9 Hessian**   QM9 Hessian (Williams et al., 2025) is a dataset of 41,645 equilibrium configurations from the QM9 dataset with numerical Hessians at the $\omega$B97x/6-31G* level. Molecular Hessians were calculated in an implicit solvation model for water, tetrahydrofuran (THF), toluene, and in vacuum. Like the baseline, all models are trained end-to-end for each solvent separately. We train on one L40s GPU for one day for HIP and three days for AD Hessian variants. The training setup is listed in table 11. The results in table 12 show that HIP improves over all baselines, in particular the Nequip-based model from Williams et al. (2025). EquiformerV2 models trained with AD Hessians via HVPs perform poorly even after three times the training wall time of HIP. Training without Hessians (E-F) yields poor accuracy on Hessian metrics.

## A.6   EXPERIMENTAL DETAILS

### A.6.1   RELAXATIONS

We compare our RFO-based optimization (Besalú & Bofill, 1998) using predicted Hessians (RFO predicted), and our RFO with BFGS updates with initial predicted Hessian (RFO-BFGS predicted init) against first-order methods, quasi-second-order BFGS variants, and exact second-order approaches obtained via automatic differentiation or finite differences.
Our baselines include steepest descent with backtracking line search (SteepestDescent) and the first-order method FIRE (Bitzek et al., 2006). For quasi-second-order methods, we consider RFO with

Table 10: **HORM-RGD1** Comparison of Hessian prediction errors on the HORM-RGD1 validation set.

| Hessian | Model | Hessian ↓ eV/Å² | Eigenvalues ↓ eV/Å² | CosSim $v_1$ ↑ unitless | $\lambda_1$ ↓ eV/Å² | Time ↓ ms |
|---------|-------|--------|-------------|---------------|------|------|
| | AlphaNet | 0.456 | 4.008 | 0.734 | 0.545 | 935.4 |
| | LEFTNet-CF | 0.243 | 1.388 | 0.821 | 0.337 | 1358.9 |
| AD | LEFTNet-DF | 0.292 | 1.319 | 0.809 | 0.301 | 434.3 |
| | EquiformerV2 | 0.092 | 0.315 | 0.822 | 0.220 | 946.1 |
| | EquiformerV2 (E-F) | 1.234 | 10.349 | 0.397 | 0.969 | 946.7 |
| Predicted | HIP-EquiformerV2 | 0.057 | 0.158 | 0.943 | 0.130 | 50.9 |
| | HIP-EquiformerV2 (end-to-end) | **0.049** | **0.123** | **0.950** | **0.098** | **40.6** |

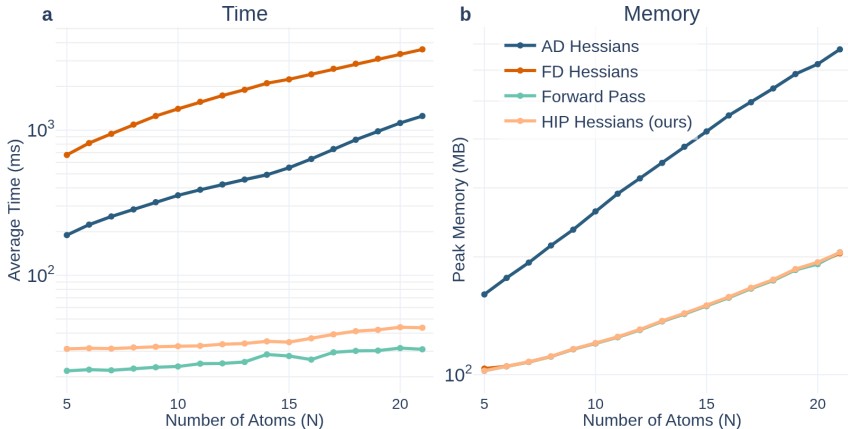

Figure 8: **Computational cost.** Comparing (a) time and (b) memory of a forward without Hessians, with HIP, automatic differentiation (AD), finite difference Hessians. The memory for Forward Pass, FD Hessians, and HIP Hessians is nearly identical.

| Model | Hessian Columns | Batch Size | Train Time / Epoch (s) |
|---|---|---|---|
| EquiformerV2 (E-F) | 0 | 64 | 380 |
| EquiformerV2 | 2 | 16 | 1900 |
| EquiformerV2-CF | 1 | 12 | 3200 |
| HIP-EquiformerV2 | 3N | 64 | 550 |

Table 11: **Training configuration and epoch time for QM9 Hessian.** Training was performed on a L40s GPU. Batch size is the maximum that could be fitted on the GPU.

BFGS updates initialized from either the identity (RFO-BFGS unit init), AD Hessian (RFO-BFGS AD init), or finite difference Hessian (RFO-BFGS FiniteDifference init). For full second-order methods, we include RFO with either automatic differentiation or finite difference Hessians. We run relaxations on 80 reactant geometries from the Transition1x validation set (which is different from HORM-T1x), that we noise with 0.5 Å RMS. We use the RS-RFO and redundant coordinate implementation in pysisyphus (Steinmetzer et al., 2021; Herbol et al., 2017). We set the convergence criteria to "Gaussian default" (see A.6.4), with a budget of 150 steps.

### A.6.2 TRANSITION STATE SEARCH WITH REACTBENCH

We use the recently introduced ReactBench benchmark (Zhao et al., 2025). The workflow consists of generating an initial guess using the growing string method (Aldaz et al., 2025; Zimmerman, 2015), followed by local search using the restricted-step partitioned rational-function-optimization method (RS-P-RFO) (Besalú & Bofill, 1998). Finally, convergence to the correct transition state is confirmed by frequency analysis, and following the intrinsic reaction coordinate (IRC) (Steinmetzer et al., 2021).

Following previous work (Zhao et al., 2025; Cui et al., 2025), we report success metrics of each step:

1. GSM successfully converged below a force RMS of $5e^{-5}$ Hartree/Bohr within 100 iterations ("GSM Success")

2. After RS-P-RFO, the frequency analysis determines geometry as a true transition state ("TS Success")

3. IRC converged to the same criteria and yields the original initial reactant and product ("IRC Verified").

We then treat the samples successfully passing (a)-(c) as transition state proposals, and verify for a random subset of 100, if the DFT Hessians have one negative eigenvalue and the DFT force RMS is below $2e^{-3}$ Ha/Bohr. We additionally report if RS-P-RFO converged to "Gaussian default" (A.6.4) within 50 steps.

Table 12: **Comparison of Hessian prediction errors on the QM9 Hessian dataset.** Denoted are the mean absolute error (MAE), cosine similarity (CosSim), and average time per forward pass on a L40s GPU. $\lambda_1$ and $v_1$ where obtained after Eckart-projection. Bold values highlight the best-performing model. E3x-Neuqip models are taken from Williams et al. (2025).

| Solvent | Model | Hessian | Forces $\downarrow$ eV/Å | Hessian $\downarrow$ eV/Å$^2$ | Eigenvalues $\downarrow$ eV/Å$^2$ | CosSim $v_1$ $\uparrow$ unitless | $\lambda_1$ $\downarrow$ eV/Å$^2$ |
|---|---|---|---|---|---|---|---|
| | E3x-Nequip | AD | 0.035 | 0.070 | - | - | - |
| | EquiformerV2 | AD | 0.030 | 0.340 | 2.641 | 0.425 | 0.151 |
| Vacuum | EquiformerV2 (E-F) | AD | 0.002 | 2.107 | 32.471 | 0.076 | 0.175 |
| | EquiformerV2-CF | AD | 0.049 | 0.881 | 12.212 | 0.074 | 0.652 |
| | HIP-EquiformerV2 (end-to-end) | Predicted | **0.002** | **0.040** | **0.172** | **0.975** | **0.010** |
| | E3x-Nequip | AD | 0.028 | 0.067 | - | - | - |
| | EquiformerV2 | AD | 0.030 | 0.281 | 1.874 | 0.380 | 0.117 |
| THF | EquiformerV2 (E-F) | AD | 0.002 | 2.077 | 31.566 | 0.067 | 0.169 |
| | EquiformerV2-CF | AD | 0.048 | 0.846 | 10.447 | 0.079 | 0.523 |
| | HIP-EquiformerV2 (end-to-end) | Predicted | **0.002** | **0.060** | **0.240** | **0.878** | **0.033** |
| | E3x-Nequip | AD | 0.029 | 0.062 | - | - | - |
| | EquiformerV2 | AD | 0.030 | 0.320 | 2.232 | 0.424 | 0.104 |
| Toluene | EquiformerV2 (E-F) | AD | 0.002 | 2.095 | 31.606 | 0.072 | 0.158 |
| | EquiformerV2-CF | AD | 0.361 | 4.794 | 39.822 | 0.023 | 320.933 |
| | HIP-EquiformerV2 (end-to-end) | Predicted | **0.002** | **0.038** | **0.117** | **0.947** | **0.008** |
| | E3x-Nequip | AD | 0.028 | 0.054 | - | - | - |
| | EquiformerV2 | AD | 0.028 | 0.273 | 1.817 | 0.377 | 0.071 |
| Water | EquiformerV2 (E-F) | AD | 0.001 | 2.097 | 31.862 | 0.066 | 0.050 |
| | EquiformerV2-CF | AD | 0.068 | 0.991 | 11.440 | 0.070 | 1.290 |
| | HIP-EquiformerV2 (end-to-end) | Predicted | **0.001** | **0.025** | **0.091** | **0.926** | **0.012** |

| Setting | Max Force | RMS Force | Max Step | RMS Step | Used for |
|---|---|---|---|---|---|
| Gaussian loose | 1.7e-3 | 1.0e-2 | 6.7e-3 | | |
| Gaussian default | 4.5e-4 | 3.0e-4 | 1.8e-3 | 1.2e-3 | Relaxation, TS search 4 |
| Gaussian tight | 1.5e-5 | 1.0e-5 | 6.0e-5 | 4.0e-5 | ZPE 4 |
| Gaussian very tight | 1.0e-6 | | 6.0e-6 | 4.0e-6 | |

Table 13: Convergence metrics used in our experiments

### A.6.3 ZERO-POINT ENERGY

Starting from 80 reactant and product geometries from the Transition1x validation set, we relax the geometries using DFT and RS-RFO (BFGS, identity initialization), up to "Gaussian tight" convergence (see A.6.4). We use the same level of theory for the relaxation as for the training of the models. Then we compute the ZPE from the Eckart-projected, mass-weighted Hessian as predicted by the different models and compare to DFT. We report all energies in eV per molecule.

### A.6.4 CONVERGENCE CRITERIA

Throughout our experiments, we adopt the following convergence criteria, set forth by the Gaussian software and widely used in different codebases. All criteria need to be met for a geometry to be considered converged.

MASS-WEIGHTING AND ECKART PROJECTION

We briefly describe the process of removing the redundant degrees of freedom of the Hessian (Wilson et al., 1955; Louck & Galbraith, 1976).

We start from a system of $N$ atoms with Cartesian coordinates $\mathbf{q} = (x_1, y_1, z_1, \ldots, x_N, y_N, z_N)^\top \in \mathbb{R}^{3N}$, atomic positions $\mathbf{R}_i = (x_i, y_i, z_i)^\top$, and masses $m_i$.

**Centering at the center of mass** The first step is to use COM-centered coordinates $\mathbf{r}_i = \mathbf{R}_i - \mathbf{R}_{\text{com}}$, which are computed from the total mass $M = \sum_{i=1}^{N} m_i$, and the center of mass (COM)

$$\mathbf{R}_{\text{com}} = \frac{1}{M} \sum_{i=1}^{N} m_i \, \mathbf{R}_i. \tag{46}$$

**Mass-weighing**  Given a Cartesian Hessian $\mathbf{H} \in \mathbb{R}^{3N \times 3N}$, its mass-weighted form is

$$\widetilde{\mathbf{H}} = \mathbf{M}^{-1/2}\, \mathbf{H}\, \mathbf{M}^{-1/2}. \tag{47}$$

where the diagonal mass matrix is

$$\mathbf{M} = \mathrm{diag}\big(\underbrace{m_1, m_1, m_1}_{\text{atom 1}}, \ldots, \underbrace{m_N, m_N, m_N}_{\text{atom } N}\big) \in \mathbb{R}^{3N \times 3N}. \tag{48}$$

**Inertia tensor and principal axes**  To build the Eckart vectors, we require the eigendecomposition of the inertia tensor. The inertia tensor about the COM is

$$\mathbf{I} = \sum_{i=1}^{N} m_i\Big[(\mathbf{r}_i \cdot \mathbf{r}_i)\, \mathbf{1}_3 - \mathbf{r}_i\, \mathbf{r}_i^\top\Big] = \begin{pmatrix} I_{xx} & I_{xy} & I_{xz} \\ I_{xy} & I_{yy} & I_{yz} \\ I_{xz} & I_{yz} & I_{zz} \end{pmatrix}, \tag{49}$$

with components

$$I_{xx} = \sum_i m_i(y_i^2 + z_i^2), \quad I_{yy} = \sum_i m_i(x_i^2 + z_i^2), \quad I_{zz} = \sum_i m_i(x_i^2 + y_i^2),$$
$$I_{xy} = -\sum_i m_i x_i y_i, \quad I_{xz} = -\sum_i m_i x_i z_i, \quad I_{yz} = -\sum_i m_i y_i z_i. \tag{50}$$

The eigendecomposition of the inertia tensor is then $\mathbf{I}\,\mathbf{E} = \mathbf{E}\,\mathbf{\Lambda}$, where the columns $\{\hat{\mathbf{i}}_1, \hat{\mathbf{i}}_2, \hat{\mathbf{i}}_3\}$ of $\mathbf{E}$ are orthonormal principal axes.

**Translational and rotational subspace (Eckart vectors)**  We can now define the three mass-weighted translational unit vectors $\{\mathbf{t}_\alpha\}_{\alpha=x,y,z} \in \mathbb{R}^{3N}$ (using unit vectors e.g., $\hat{\mathbf{e}}_x = (1,0,0)^\top$)

$$\mathbf{t}_\alpha^{(i)} = \sqrt{m_i}\, \hat{\mathbf{e}}_\alpha, \quad \alpha \in \{x,y,z\}, \quad i = 1, \ldots, N, \tag{51}$$

as well as the three mass-weighted rotational vectors $\{\mathbf{r}_k\}_{k=1,2,3} \in \mathbb{R}^{3N}$ using the COM-centered coordinates and the principal axes from the inertia tensor:

$$\mathbf{r}_k^{(i)} = \sqrt{m_i}\,(\hat{\mathbf{i}}_k \times \mathbf{r}_i), \quad k = 1, 2, 3. \tag{52}$$

For linear molecules, one rotational vector has (near-)zero norm and is discarded; in practice, we drop any $\mathbf{r}_k$ whose norm is below a small threshold. We collect the translation and rotation vectors as rows of a matrix

$$\mathbf{T} = \begin{pmatrix} \mathbf{t}_x^\top \\ \mathbf{t}_y^\top \\ \mathbf{t}_z^\top \\ \mathbf{r}_1^\top \\ \mathbf{r}_2^\top \\ \mathbf{r}_3^\top \end{pmatrix} \in \mathbb{R}^{d \times 3N}, \tag{53}$$

with $d = 6$ for non-linear molecules and $d = 5$ for linear molecules. We then orthonormalize these rows (e.g., QR factorization of $\mathbf{T}^\top$) to obtain an orthonormal set $\{\mathbf{u}_a\}_{a=1}^d$ spanning the rigid-body subspace in mass-weighted coordinates.

**Eckart projector**  To build the projector, two equivalent forms are convenient in practice.
1) Explicit projector

$$\mathbf{P} = \mathbf{I}_{3N} - \sum_{a=1}^{d} \mathbf{u}_a\, \mathbf{u}_a^\top. \tag{54}$$

2) Nullspace basis via the SVD $\mathbf{T}^\top = \mathbf{U}\,\mathbf{\Sigma}\,\mathbf{V}^\top$, which we use as default due to improved numerics. The columns of $\mathbf{U}$ associated with zero singular values span the vibrational subspace. Stack them row-wise to form $\mathbf{P}_{\text{basis}} \in \mathbb{R}^{(3N-d) \times 3N}$.
Finally, we apply the Eckart projection

$$\widetilde{\mathbf{H}}_{\text{Eckart}} = \mathbf{P}_{\text{basis}}\, \widetilde{\mathbf{H}}\, \mathbf{P}_{\text{basis}}^\top. \tag{55}$$

followed by re-enforcing the projected Hessian to be symmetric:

$$\widetilde{\mathbf{H}}_{\text{Eckart}} \leftarrow \tfrac{1}{2}\big(\widetilde{\mathbf{H}}_{\text{Eckart}} + \widetilde{\mathbf{H}}_{\text{Eckart}}^\top\big). \tag{56}$$

INTRINSIC REACTION COORDINATE WITH EULER PREDICTOR-CORRECTOR

The intrinsic reaction coordinate (IRC) traces the steepest-descent path on the potential energy surface (PES) from a transition state (TS) toward reactant and product minima. We use the IRC provided in the ReactBench benchmark (Zhao et al., 2025), which itself uses the pysisyphus implementation (Steinmetzer et al., 2021). We integrate in mass-weighted Cartesian coordinates to obtain a physically meaningful path, solving:

$$\frac{d\mathbf{x}}{ds} = -\frac{\mathbf{g}(\mathbf{x})}{\|\mathbf{g}(\mathbf{x})\|},$$

where $\mathbf{x}$ are mass-weighted coordinates and $\mathbf{g}$ is the mass-weighted gradient. The pysisyphus implementation follows an Euler predictor-corrector (EulerPC) scheme (Hratchian et al., 2010). Starting from a TS, we diagonalize the mass-weighted Hessian, identify the imaginary mode (transition vector), and displace slightly along it to initialize the path. We perform small Euler integrations along $-\mathbf{g}/\|\mathbf{g}\|$, updating the gradient via a local Taylor expansion using the current Hessian. The Hessian is updated with BFGS updates from $\Delta\mathbf{x}, \Delta\mathbf{g}$. We refine the predicted point using distance-weighted interpolation (DWI) of previously visited points (coordinates, energies, gradients, Hessians). The corrector integrates the IRC equation on this surrogate PES using a modified Bulirsch-Stoer procedure with Richardson extrapolation.

