# OpenReview forum: "Shoot from the HIP: Hessian Interatomic Potentials without derivatives"
_ICLR.cc/2026/Conference — Submitted to ICLR 2026_

### Official Review · Reviewer_C1zb · 2025-10-29

**Soundness:** 3
**Presentation:** 3
**Contribution:** 3
**Rating:** 2
**Confidence:** 4

**Summary:**

In this paper, the authors investigate the accuracy of directly predicting molecular Hessian, rather than deriving them from numerical derivatives of forces or automatic differentiation, for predicting vibrational properties of molecules. They claim that forward prediction is “one to two orders of magnitude faster, more accurate, more memory efficient, easier to train, and enables more favorable scaling with system size” compared to autodifferentiation. For evaluation, the authors use an EquiformerV2 backbone to predict Hessians of the HORM dataset of organic molecules and compare HIP-EquiformerV2 against AlphaNet, LEFTNet, LEFTNet-df, and EquiformerV2 using autodifferentiation.

This paper addresses an important bottleneck in atomistic simulation. Benchmarks of this kind are valuable for the community, as they clarify the tradeoffs between direct prediction and differentiation-based approaches and help researchers decide which method is most appropriate for a given use case. The experiments are promising, but the claims about speed and generality need expanded benchmarking and clearer implementation transparency to be a solid resource for the community and thus merit acceptance.

**Strengths:**

* The paper is well written, accessible, and gives a solid overview of relevant literature, including clear derivations of how Hessians can be constructed from edge features using Clebsch–Gordan coefficients.
* The experimental evaluation is comprehensive, comparing multiple aspects of Hessian prediction: convergence steps, wall time, accuracy of geometry optimization, zero-point energy, transition-state search, frequency analysis, and raw elementwise accuracy.
* The authors clearly state limitations, such as reliance on ground-truth DFT data and the current restriction to small molecular systems.

**Weaknesses:**

* The direct prediction of Hessians is not, on its own, a sufficiently novel contribution to carry the paper. Its impact depends on demonstrating clear and reproducible efficiency–accuracy tradeoffs across architectures and datasets, which currently feels underexplored.
* It is unclear how much of the difference in AD vs. HIP accuracy for EquiformerV2 arises from training setup and task formulation since EquiformerV2 was not originally designed or tuned for Hessian prediction vs. being a “fundamentally” better approach to learn second derivatives directly. The authors should discuss how architectural or training adaptations might influence these results, and whether the comparison reflects representational capacity or differing task suitability.
* While numerical derivatives scale poorly, they remain heavily used due to simplicity and established workflows. A direct wall-time comparison (finite-difference forces vs. HIP vs. AD) would contextualize the claimed speedups.
* The comparisons of memory, batching, and wall time are highly implementation-dependent. The paper would benefit from a reproducible table specifying hardware, batch size, precision, and AD implementation details (e.g., JAX vs. PyTorch) since these factors dominate observed scaling.
* When training Hessians via AD, one would typically not predict full Hessians every iteration but use sampled Hessian-vector products or partial training, which can substantially mitigate costs. This nuance should be acknowledged to make the efficiency discussion fairer.
* As far as I can tell, the HORM benchmark is not peer-reviewed. Including results on an established dataset (e.g., Hessian-QM9) or a small material-system test would strengthen generality claims.
* The paper evaluates HIP vs. AD only for EquiformerV2. It would be informative to test lighter-weight architectures (e.g., NequIP, MACE, Allegro) to see whether the efficiency and accuracy gaps persist across model families.

**Questions:**

1. Could the authors clarify the exact computational setup for timing and memory benchmarks—hardware type, precision, AD backend (JAX vs. PyTorch), and batch sizes? These details are critical for reproducibility.
2. How sensitive are the reported speed and memory gains to the choice of AD implementation or backend?
3. Given that only the Hessian head was retrained on a frozen backbone, do the authors expect joint end-to-end training of energies, forces, and Hessians to change the relative trends between HIP and AD?
4. Would the HIP approach generalize straightforwardly to periodic or larger material systems, or are there challenges related to locality assumptions and cutoff sparsity?

---

> ### Author Response · Authors · 2025-11-20
>
> > How much of the difference in AD vs. HIP accuracy for EquiformerV2 arises from training setup ... should discuss how architectural or training adaptations might influence these results...
>
> If we understand your concern, you are asking whether EquiformerV2 is inherently limited in its ability to produce accurate AD-Hessians, and whether a model designed specifically for AD based Hessian prediction might narrow the gap with HIP. We agree that this is an interesting and important question.
>
> However, the HORM study represents the most comprehensive exploration of Hessian predictions with MLIPs, and EquiformerV2 is currently the strongest performing model. Our comparison therefore reflects the current state of the art.
>
> Importantly, all AD baselines were trained
> 1) on ground-truth DFT Hessians
> 2) with significantly larger compute budgets than HIP
>
> Even if future architectures improve AD-Hessians, HIP would remain fundamentally faster at inference; the core advantage of our work.
>
> >A direct wall-time comparison (finite-difference forces vs. HIP vs. AD) would contextualize speedups.
>
> Please see EquiformerV2 timings von Transition1x data below, (FD=finite differences), RTX 3060
> |Type|Time [ms]|
> |---|---|
> |AD from AD forces|1434|
> |AD from direct forces|576|
> |FD from AD forces|3464|
> |FD from direct forces|2067|
> |HIP|31|
>
> >The paper would benefit from a table specifying hardware, batch size, precision, and AD implementation details (e.g., JAX vs. PyTorch)
>
> We will add missing details in our hyperparameter table. Adressing each point individually:
> - We used RTX3060 GPUs for inference of all models. For training we used H100
> - Inference batch size=1. Training batch size=128
> - Precision: float32
> - All AD implementations are from [1] and here: https://github.com/deepprinciple/HORM
> - Software is PyTorch. While Jax AD is often faster, the forward and therefore HIP would also be faster. The HVP implementations in Jax and PyTorch are fundamentally similar with same theoretical scaling, see [2]. The basic limitation of AD Hessians is that they require $3N_\text{Atoms}$ HVPs. Each HVP is minimum ~3x the cost of the forward. To demonstrate, we calculate AD Hessians of Aspirin (63x63 dimensional) in the JAX implementation of MACE and compare to the cost of a single MACE forward pass:
>
> |Method|Cost Hessian/Cost of single forward|
> |---|---|
> |For-Loop (Jax)|166.91x|
> |Vmap (Jax)|40.26x|
> |HIP|1.3x|
>
> The cost of the for-loop agrees with theory. Vmaping HVPs is 4x faster, while consuming much more memory. HIP only needs 1.3x the time and the same memory as the forward.
>
> Are there any other settings you suspect could change our conclusions?
>
> >When training Hessians via AD, one would typically not predict full Hessians every iteration but use sampled Hessian-vector products
>
> We agree and the baseline models from [1] are already trained with this HVP protocol. Despite this, training remains exceedingly expensive. The HORM paper states that they use 2 HVPs at a batch size of 128, and their checkpoint metadata shows that the per device batch size was 8, implying a 16xA30 GPU setup for 1 million gradient steps.
> In contrast, we trained HIP on a single H100 for 500k steps. Each train step of HIP costs 1/5 the cost of AD Equiformer training on our hardware.
>
> >Including results on an established dataset (e.g., Hessian-QM9)
>
> We trained three Equiformer models on QM9. HIP is trained for 1 day, AD for 3 days. The water solvent split results are below, the full results in the pdf.
>
> |Model|MAE (meV/A^2)|
> |---|---|
> |NequIP from QM9 authors|0.054|
> |AD from AD forces|2.118|
> |AD from direct forces|0.273|
> |HIP|0.025|
>
> Equiformer's AD Hessian from AD forces is so slow that 3 days was not enough train time. NequIP is also AD from AD forces trained to convergence and still has lower accuracy than HIP
>
> >It would be informative to test lighter-weight architectures (e.g. NequIP) to see whether the efficiency and accuracy gaps persist
>
> Given the high training costs of AD Hessian it is unfortunately infeasible to train another model during the rebuttal. The efficiency gap wont be different though, given AD scaling is $3N_{atoms}$x vs HIP's 1x forward pass.
>
> >do the authors expect joint end-to-end training change the relative trends between HIP and AD?
>
> We retrained HIP end-to-end and found that it further improves the Hessians
>
> |Method|Hessian MAE eV/A^2|
> |---|---|
> |Frozen backbone|0.03|
> |Joint training|0.02|
>
> >Would HIP generalize to periodic or large material systems? Challenges related to locality assumptions?
>
> Periodicity is supported by the model: We can just average the messages over periodic images.
> From a physical point of view, any challenge in non-locality of the Hessian would also be a challenge in the energy. Therefore, assuming the MLIP can model the energy, HIP can also model the Hessian
>
> [1] arXiv preprint arXiv:2505.12447 (2025).
>
> [2] https://iclr-blogposts.github.io/2024/blog/bench-hvp

---

> > ### Comment · Reviewer_C1zb · 2025-11-26
> >
> > Thank you for your responses to my questions.
> >
> > A weakness of the paper that I think will be impossible to resolve this iteration is that that comparisons are being made to methods that were not optimized or published with Hessians in mind, as this would have likely driven different design decisions by the original authors. Additionally, the paper's results would be much more useful if a "best" attempt was made on all fronts to make the comparison -- i.e. what's the fastest and most accurate method you can build using autodiff Hessians vs. numerical difference Hessian via forces vs. direct prediction and generally how does this impact the energy landscape learned in terms of smoothness. I recognize that this a huge amount of effort, but it really would be the most useful paper to the community.
> >
> > What can be done this iteration -- I should've formulated these also as questions, but I'd appreciate if the authors could also address / discuss the weakness I brought up in the review. In particular, it would be interesting to compare numerical derivatives vs. directly predicted Hessians in terms of speed but more importantly whether they converge to the same answer for the same model -- indicative of whether the forces are Hessian are representing compatible energy surfaces.
> >
> > Regarding the author's response about applying their method to extended systems, it's important to note that Hessians for extended systems are formulated using the Dynamical Matrix which includes a dependence on crystal momentum $\vec{q}$ -- essentially, this addresses the energetic cost of translation symmetry breaking distortions. One need to add a phase factor dependent on cell image atomic position and $\vec{q}$ to each "edge" before averaging to achieve the final $\vec{q}$ dependent Hessian. [You can find a quick background here,](https://www.vasp.at/wiki/Phonons:_Theory#:~:text=We%20can%20write%20the%20positions,atoms%20in%20the%20unit%20cell.) but Ashcroft and Mermin Ch 22 -- specifically Eqn. 22.58 is a more comprehensive resource.

---

> > > ### Author Response · Authors · 2025-11-27
> > >
> > > ## "Best" attempt: What's the fastest and most accurate method you can build using autodiff Hessians vs. numerical difference Hessian via forces vs. direct prediction
> > >
> > > All baselines were already purposefully trained for AD Hessians.
> > > We compare to the strongest baselines provided in the largest systematic Hessian study to date (HORM), where the authors already reported results for state-of-the-art architectures and tuned the hyperparameters. Previous papers attempted to improve numerical and AD Hessians with different training strategies, on different datasets, and with different architectures to some success [1,2,3], but the resulting Hessian errors were always comparable or worse to the ones in HORM.
> > >
> > > It is not clear to us how a "best attempt" could be meaningfully defined or implemented in practice beyond what we did.
> > > Designing new baselines beyond existing work goes substantially beyond the scope of typical benchmarking in MLIP works and would introduce a significant amount of ambiguity regarding what constitutes a "optimal" design.
> > >
> > > Even if one were to succeed in greatly optimizing the baselines, the fundamental scaling differences remain unchanged. Direct Hessian prediction has $O(1)$ whereas both autodiff-based and finite-difference Hessians inherently require $O(N)$ costs in terms of forward passes. Because these costs arise from algorithmic structure rather than architecture or hyperparameter choices, no amount of tuning is expected to close this gap.
> > > Therefore, if accurate autodiff or numerical Hessians can be obtained in the future, one can use them to generate Hessian labels. These can then easily be "distilled" into a direct prediction head, to benefit from the fundamentally better speed and scaling of our direct prediction.
> > >
> > > [1] https://www.nature.com/articles/s41467-024-52481-5
> > >
> > > [2] https://pubs.acs.org/doi/10.1021/acs.jpcc.4c07477
> > >
> > > [3] https://pubs.acs.org/doi/10.1021/acs.jctc.5c00402
> > >
> > > ## Compare numerical derivatives vs. directly predicted Hessians in terms of speed
> > >
> > > Thank you for highlighting this again, so we can clarify where we include these results.
> > >
> > > For the speed comparison to finite difference Hessian please see the first table in our previous response.
> > > We also included speed and memory scaling comparisons between numerical Hessians and HIP in the PDF in figure 8.
> > > HIP is consistently one to two orders of magnitude faster and more memory efficient, even for small systems.
> > > If this does not answer the question, can you pinpoint what speed comparison you mean?
> > >
> > > ## Whether numerical derivatives vs. directly predicted Hessians converge to the same answer for the same model and forces and Hessians are compatible
> > >
> > > By "converging", do you mean the geometry optimization and transition state search tasks?
> > >
> > > HIP is trained to predict the DFT Hessians, not the model's AD/FD Hessians. We therefore only expect HIP to predict the same answer as AD/FD, if the model's AD/FD Hessian approximate the DFT Hessians well.
> > >
> > > For geometry optimization: The Hessian is used as a preconditioner in RFO. A positive definite preconditioner does never change the minima, only the number of steps until convergence. It can theoretically change which minima is reached if several a closeby, but this is true for any preconditioner, including the widely used BFGS Hessian approximation or the true Hessian itself.
> > >
> > > For transition state search: We verify with DFT that the MLIP transition states are true transition states, and HIP finds more true transition states than AD. We also verify that the relaxations from the transition state falls back down to the reactant and product states we started from and again, HIP finds more of these "IRC verified" transition states. Since we find more TS with HIP than with AD, it is not informative to compare if we find the same.
> > > We can therefore conclude that the HIP Hessians are very good local quadratic surrogates of the PES.
> > >
> > > If we really want the HIP Hessian to match the AD/FD Hessian, we would train HIP on those AD/FD Hessian labels instead of the DFT Hessian labels, to distill the Hessians and predict them much faster during inference.
> > >
> > > ## Dynamical Matrix for extended systems
> > >
> > > Thank you for the question. You are right that the dynamical matrix is momentum dependent.
> > > We were talking about the periodic Hessian of a supercell, which we can obtain in a HIP style. The cartesian supercell Hessian can then be Fourier transformed into the dynamical matrix (which induces the momentum dependency), which is a closed form, cheap operation.
> > > This is the same protocol used in existing works that start from MLIP-AD Hessians and convert them into a dynamical matrix for phonon calculations [1]; in our case HIP would simply provide the real-space Hessian directly instead of via AD.
> > >
> > > [1] https://arxiv.org/html/2403.11347v1#S7.F1

---

### Official Review · Reviewer_22z7 · 2025-10-31

**Soundness:** 3
**Presentation:** 3
**Contribution:** 1
**Rating:** 4
**Confidence:** 4

**Summary:**

The paper introduces the Hessian Interatomic Potential (HIP) framework, which leverages SE(3)-equivariant neural networks to directly predict the full (3N \times 3N) molecular Hessian matrix, thereby accelerating the computation of second-derivative information in molecular simulations. In terms of architecture, HIP employs EquiformerV2 as the core equivariant backbone, coupled with a Clebsch–Gordan (CG) expansion–based prediction head to construct the final Hessian tensor with proper rotational symmetry. Furthermore, the method incorporates a loss function that includes eigenvector-based constraints, which guide the model to produce Hessian predictions with more accurate eigenvalue spectra, improving the quality of vibrational and stability properties derived from the predicted matrices.

**Strengths:**

1. The paper aims to provide a comprehensive evaluation of the predicted Hessian matrices through multiple downstream tasks, including geometry optimization, zero-point energy (ZPE) estimation, and transition-state (TS) searches to test the accuracy and the traditional methods.

**Weaknesses:**

1. The paper lacks sufficient discussion and comparison with existing SE(3)-equivariant networks designed for predicting equivariant matrices. A series of works following PhiSNet have explored predicting the Hamiltonian matrix, which is also an equivariant matrix similar in nature to the Hessian. The architectures developed in those studies could, in principle, be applied to Hessian prediction as well. Therefore, including an introduction and comparison with these related models would help demonstrate the advantages of the proposed approach in this prediction task.

2. Furthermore, regarding the loss function design, paper [1] introduces a method that imposes constraints on the eigen-energies of the Hamiltonian matrix. The loss function presented in Section 3.3 of the current submission appears to be identical to that in [1], and this overlap should be clearly acknowledged and discussed.

3. Overall, the writing and organization of the paper are clear. However, my main concern lies in the novelty of the machine learning contribution. Most of the techniques employed have already been introduced in prior works, and the Hessian matrix prediction task closely resembles Hamiltonian matrix prediction in its formulation—an area that has been previously studied but not sufficiently discussed or differentiated in this submission.

**Questions:**

N/A.

---

> ### Author Response · Authors · 2025-11-20
>
> > 1. The paper lacks sufficient discussion and comparison with existing SE(3)-equivariant networks designed for predicting equivariant matrices...
>
> Thank you for the feedback and for highlighting the connections to the rich field of Hamiltonian prediction. We have included a discussion of these models and tasks in the introduction. We agree that some of the design decisions introduced in these papers might be useful for Hessian prediction as well. Maybe you have some specific techniques in mind?
> We want to stress that the goal in our paper is not to find the best possible architecture for equivariant matrix prediction, but to establish direct Hessian prediction as a interesting ML task and viable technique by demonstrating the advantages and validating practical relevance on extensive downstream benchmarks.
>
> > 2. Furthermore, regarding the loss function design, paper [1] introduces a method that imposes constraints on the eigen-energies of the Hamiltonian matrix...
>
> The paper is not linked here, but you probably mean the WALoss paper [1]? We have found the work since submission and agree that the loss is similar in spirit and mainly differ in application details; we will clarify this in our paper.
>
> >3. Overall, the writing and organization of the paper are clear. However, my main concern lies in the novelty of the machine learning contribution...
>
> We value the reviewer's feedback that the application of techniques similar to Hamiltonian prediction may initially appear to have limited novelty. We would like to politely push back against this perspective, as our framework and design choices fundamentally challenge the predominant approach of obtaining nuclear Hessians. The fact that we can circumvent automatic differentiation and construct equivariant Hessians from message passing is a novel insight.
>
> We further wish to hightlight the contributions of our work: (1) our problem formulation is new and has its own set of challenges. (2) There have been many works trying to obtain Hessians from MLIPs, including using Hessian labels for finetuning and exhaustive comparison of methods to obtain Hessians from MLIPs [2], yet no work trying to directly predict them. (3) We provide, for the first time, strong empirical evidence that the advantages of direct Hessian prediction translate to successfull downstream tasks.
>
> Following ICLR’s reviewer guidelines, originality does not require inventing an entirely new architectural primitive;
> The official ICLR reviewing instructions state that "originality may arise from a new definition or problem formulation, creative combinations of existing ideas [and] application to a new domain...".
> This is precisely the aim of our work: We take an idea that was not previously recognized as applicable to Hessians, adapt it, and rigorously validate that it provides large accuracy and efficiency benefits.
>
> We hope that the reviewer may join us in agreeing that scientific value can not only be created by novel architectures, but also by bringing techniques to new tasks and rigorously applying and testing them and show their practical usefulness.
>
> [1] Li, Yunyang, et al. "Enhancing the Scalability and Applicability of Kohn-Sham Hamiltonians for Molecular Systems" arXiv preprint arXiv:2502.19227 (2025).
>
> [2] Cui, Taoyong, et al. "Horm: A large scale molecular hessian database for optimizing reactive machine learning interatomic potentials." arXiv preprint arXiv:2505.12447 (2025).

---

### Official Review · Reviewer_vyqg · 2025-11-01

**Soundness:** 2
**Presentation:** 3
**Contribution:** 2
**Rating:** 4
**Confidence:** 2

**Summary:**

This paper proposes a derivative-free method to predict Hessian matrices for interatomic potential models, which reduces the computational cost of Hessian evaluation. The method directly constructs SE(3) equivariant symmetric Hessians from higher-order equivariant features of a graph neural network. This lowers both memory and time costs for training and inference that involve Hessians in interatomic potentials. The paper validates the trained models on multiple downstream tasks and the accuracy remains consistently strong.

**Strengths:**

1.It proposes a novel derivative-free Hessian prediction method and a loss function tailored to the Hessian subspace.
2.The Hessian prediction method guarantees equivariance and symmetry, and the approach is simple, reasonable, and self-consistent.
3.The method accelerates Hessian prediction, reducing complexity from O(N^2) to O(N), and it also lowers memory usage.
4.The method achieves high accuracy and performs well across various downstream tasks.

**Weaknesses:**

1.Direct Hessian prediction is not applicable when DFT Hessian data are lacking.
2.The reliability of direct Hessian prediction is uncertain.
3. This method will be quite likely to fail when long-time simulation is required.

**Questions:**

1.Obtaining Hessian data from DFT is difficult, and there are currently few DFT datasets that include Hessians. In this context, AD-based Hessian prediction can produce Hessians without training specifically on Hessians. In contrast, direct Hessian prediction necessarily requires Hessian data for training, which limits its applicability.
2.The generated Hessian matrices satisfy rotational equivariance and symmetry. However, direct Hessian prediction does not arise from differentiating energy or forces. Could this lead to other physical inconsistencies? For example, physically there should be six zero eigenvalues. Although this is included in the loss, it is not guaranteed by the model architecture. The paper provides extensive numerical and downstream evidence for effectiveness, and direct Hessians do not suffer from the non-conservative force issue. Even so, this potential non-physical aspect remains concerning. Could the authors offer theoretical discussion and clarification?
3.In Table 1, the Hessian accuracy of the AD baselines is similar to that reported in the HORM paper, but the eigenvalue accuracy differs considerably. Was a different error metric used?
4.The authors claim improved Hessian prediction accuracy. With Equiformer V2 as the backbone and only training a separate Hessian head, improvements are indeed observed, even in loss ablations. However, the Equiformer V2 backbone comes from HORM and has already been trained with Hessian data. This work then trains on that backbone. The authors state this isolates the effects of energy and forces, but this seems unfair. Could the authors provide results where the backbone is trained only on energy and forces, then frozen, and the Hessian head is trained on top?
5.Among the AD baselines, some models use direct forces. Even with AD, the resulting Hessians are not fully physical. The comparison between LEFTNet and LEFTNet-df shows that conservative force models yield more accurate Hessians. The AD baseline for Equiformer V2 uses direct forces. Would an AD Hessian with conservative forces on Equiformer V2 achieve higher accuracy?
6.Typos: line 102 Hesians. line 465 should be Table 3. line 1001 transtion.
7. Can this method work for long time simulation?

---

> ### Author Response · Authors · 2025-11-20
>
> >The reliability of direct Hessian prediction is uncertain.
>
> We hope the mathamatical discussion and extra experiments below addresses your concerns. If not, can you explain what extra experimental evidence would change your mind?
>
> >This method will be quite likely to fail when long-time simulation is required.
>
> Do you have any application in mind? We are not aware of any mainstream task where Hessians are integrated over a long time.
>
> >Could the authors offer theoretical discussion and clarification?
>
> We agree that it is important to understand when the non-integrability error can lead to catastrophically wrong downstream predictions, similar to direct forces in MD simulations, and that the paper would benefit from a theoretical discussion.
> We require forces to be conservative, since molecular dynamics accumulates errors scaling with integration length.
> There is no application we are aware of that integrates the Hessians like this, meaning we do not requires the Hessian to be conservative. We formalize this in what follows. Please let us know if this is helpful:
>
> Lets first review the more familiar case: Why do direct force predictions fail in MD simulations but succeed in geometry optimization? Suppose we directly predict forces with error $e(x)=F_\\text{pred}(x)-F(x)$. We can decompose the error in a conservative and a non-conservative field using the Helmholtz decomposition: $e(x)=\nabla\phi(x)+S(x)$. $\phi\in\mathbb{R}$ is a scalar potential, and $S$ is a vector field $S(x)\in\mathbb{R}^{3N}$ that is divergence-free $\nabla\cdot S(x)=0$ but not curl-free $\nabla\times S(x)\neq 0$.
> Now consider how the energy of the system changes over time in an MD simulation:
> $$
> \\begin{align}
> \\Delta E' &=\\int_0^T F_\\text{pred}(x(t)) v(t) dt = \\int_0^T (F(x(t)+e(x(t))) v(t) dt \\\\
> &=\\int_0^T (F(x)+\\nabla\\phi(x) +S(x)) v(t) dt \\\\
> &= E(T) - E(0) + \\underbrace{\\phi(T) - \\phi(0)}\_{\text{ error bounded with time}} + \\underbrace{\\int_0^T S(x) v(t) dt}\_{\\text{error unbounded with time}}
> \\end{align}
> $$
> Since MD simulations use large $T$, non-conservative forces lead to the blowups in error.
>
> In geometry optimization however, we define $E'=E+\phi$, then follow the gradient descent ODE:
> $$
> \\begin{align}
> \\frac{dE'(x(t))}{dt} &= \\nabla E'(x(t))\\cdot\\frac{d x(t)}{dt} \\\\
> &= -\\nabla E'(x(t)) \\cdot (\\nabla E(x) + e(x))\\\\
> &= -\\nabla E'(x(t)) \\cdot (\\nabla E(x) +\\nabla\\phi(x) +S(x))\\\\
> &= -\\nabla E'(x(t)) \\cdot (\\nabla E'(x) + S(x))\\\\
> &= -|\\nabla E'(x(t))|^2 +\\nabla E'(x)\\cdot S(x)
> \\end{align}
> $$
> Using the general inequality $2ab\leq a^2 +b^2$:
> $$
> \\begin{align}
> \\frac{dE'(x(t))}{dt} &= -|\\nabla E'(x(t))|^2 + \\nabla E'(x) \\cdot S(x) \\\\
> &\\leq -|\\nabla E'(x(t))|^2 + \\frac{1}{2}|\\nabla E'(x(t))|^2 + \\frac{1}{2}|S(x)|^2 \\\\
> &=-\\frac{1}{2}|\\nabla E'(x(t))|^2 +\\frac{1}{2}|S(x)|^2
> \\end{align}
> $$
> Therefore, as long as $|\nabla E'(x(t))|^2 > |S(x)|^2$, the direct predicted force minimizes the energy plus some error potential similar to AD forces. Once we are close to convergence we have $|\nabla E'(x(t))|^2 \leq |S(x)|^2$ and stop converging or enter a limit cycle. This means, we will converge to an extrema on a energy surface $E'$ within a distance of $O(|S|)$. In practice we are interested in how close we converge to the extrema of the DFT energy $E$. For this it doesnt matter if the error stems from the error potential $\phi$ shifting the extrema, or $S$ inducing a small limit cycle.
> As there is no time dependent error blow up it is therefore safe to use direct forces for geometry optimization.
>
> Now we can do the same for the directly predicted Hessian $H_\text{pred}(x)=H(x)+e(x)$:
> Denote the error again as $e(x)=\nabla^2\phi(x)+S(x)$.
> As before we would get errors scaling with integration length if we would try to calculate quantities of the form
> $$
> Q(T) = \int_0^T f(x(t),  H_\text{pred}(x(t))) dx(t)
> $$
> leading to large errors for large $T$. However, our key argument is: No mainstream task integrates over a function of the Hessian!
> ZPE and geometry classification are single point calculation, therefore the errors $\nabla^2\phi(x)$ and $S(x)$ both behave the same, one error is not "worse" than the other.
> Mode following in transition state search is equivalent to the geometry optimization analysis with direct forces above. Energy minimization with Hessian preconditioner is also unproblematic, as preconditioner cannot change the minimum, they only change the convergence speed. In fact, it is common practice to use BFGS updates for transition state search and energy minimization, and BFGS updates do not preserve energy either.

---

> > ### Author Response · Authors · 2025-11-20
> >
> > (2/2)
> >
> > > physically there should be six zero eigenvalues.
> >
> > This is a great point. For DFT Hessians SCF residuals and finite differences cause small errors in the null space, and one typical uses Eckart projection to project out the error [1]. We use the same procedure. Experimentally our model correctly learns the null space.
> >
> > [1] https://nwchemgit.github.io/Hessians-and-Vibrational-Frequencies.html#print-control
> >
> > > Obtaining Hessian data from DFT is difficult, and there are currently few DFT datasets that include Hessians...
> >
> > You are correct that we require Hessian labels, similar to the energy and force labels used for regular MLIPs. We highlight that the baselines also use Hessian labels to achieve decent accuracy for AD Hessians, as only training on forces does not seem to be enough to get accurate Hessians.
> >
> > Hessians are central for many quantum chemistry workflows. Therefore it seems likely more large scale datasets will be provided in the future.
> > To address this concern further, we benchmarked how much data we need to achieve good results. For our current model size, we see comparable errors to our fully trained model at around 500k data points.
> > Additionally we tested our model on the RGD1 holdout set and found the errors for the larger molecules to be comparable to the in-distribution set.
> > Therefore it is realistic to create large enough datasets that can be used in training well generalizing foundation type models.
> > Conversely, if we assume AD Hessians are reliable and no DFT Hessians are available, we can still distill the AD Hessian into HIP to get much faster Hessian predictions.
> >
> > >... the eigenvalue accuracy differs considerably. Was a different error metric used?
> >
> > The HORM paper reported the eigenvalues in the wrong units as acknowledged by the authors here ( https://www.kaggle.com/datasets/yunhonghan/hessian-dataset-for-optimizing-reactive-mliphorm/discussion/588376#3248107 )
> >
> >
> > >4.... Could the authors provide results where the backbone is trained only on energy and forces, then frozen, and the Hessian head is trained on top?
> >
> > As you pointed out, we restricted ourselves to only finetuning the Hessian readout head starting from the baseline, to isolate the Hessian from the energies and forces in the experiments.
> > To address the question directly, we trained two more settings:
> > 1. the backbone is trained only on energy and forces (E+F (frozen) + HIP)
> > 2. trained end-to-end with energies, forces, and direct Hessian (E+F+HIP)
> >
> > |Trainings setting|MAE (eV/A^2)|
> > |---|---|
> > |E+F (frozen) + HIP|0.032|
> > |E+F+H (frozen) + HIP (in paper)  |0.030|
> > |E+F+HIP|0.020|
> >
> > All HIP versions improve over the AD Hessian baseline, and end-to-end training of HIP in particular significantly outperforms all other methods.
> >
> >
> > > Would an AD Hessian with conservative forces on Equiformer V2 achieve higher accuracy?
> >
> > Thank you for the interesting suggestion.
> > We trained three Equiformer models including conservative forces on QM9, since it is small enough to train in the short rebuttal time. We train AD for three times longer than HIP (3 vs 1 day):
> >
> > |Model|MAE (meV/A^2)|
> > |---|---|
> > |NequIP from QM9 authors|0.054|
> > |AD from conservative forces|2.118|
> > |AD from direct forces|0.273|
> > |HIP|0.025|
> >
> > HIP clearly performs the best. AD Hessians from conservative forces are so slow that 3 days was not enough even on the tiny QM9 dataset. NequIP is also AD from conservative forces trained to convergence and still yields lower accuracy than HIP. We present the detailed results in the reuploaded pdf.
> >
> > We apologize for the long reply. We hope the additional discussion clarified the valid points raised by the reviewer. We amended the relevant parts in the paper.

---

### Author Response · Authors · 2025-11-20
**Response to everyone**

We thank all the reviewers for their time and constructive comments. In response to the feedback we ran several extra experiments that greatly improved the paper and we think might be interesting to all reviewers. The full results can be found in the reuploaded PDF and are summarized here:

- We trained HIP end-to-end without the pretrained backbone. This improved the Hessian accuracy significantly, and also improves force and energy accuracy across all data scales (table 1 and figure 5 in the PDF)
- Additional validation: We evaluate on the RGD1 datasplit with up to 35 atoms, significantly larger than the max of 25 atoms in the train set; HIP is still the best performing model and by far the fastest (table 10)
- Extra dataset: We train and evaluate on the QM9 Hessian dataset. HIP consistently beats the QM9 Hessian authors baseline (a NequIP architecture) and the AD Hessian Equiformer baselines trained by us on all splits while being faster (table 12)
- As suggested by reviewer vyqg, we included a mathematical discussion when direct predictions are safe
- Extra timings comparing AD Hessians from AD forces, AD Hessians from direct forces, finite difference Hessians from AD forces, Finite difference from direct forces and HIP, demonstrating HIP is significantly faster

---

### Meta-Review · Area_Chair_sdLn · 2026-01-06

**Summary:**

This paper proposes a derivative-free approach for directly predicting Hessian matrices in interatomic potential models with equivariant symmetry. The reviewers raise several major concerns that inform the suggested decision. These include (1) the additional requirement of supervised Hessian data, (2) substantial overlap with existing techniques for related tasks, and (3) the implementation choice for baselines.

**Reviewer Concerns:**

The rebuttal addresses some, but not all, of the reviewers’ concerns. The authors clarify that the predicted Hessians are not intended for long-time integration, but rather for single-point or local calculations, in which case prediction errors can be controlled. The authors present the results of end-to-end HIP training with further improved accuracy, which justifies the strenght of HIP.

However, the concern about substantial overlap with existing approaches (including architecture design and loss function) for predicting Hamiltonian matrices, remains largely unaddressed.

**Reviewer Scores:**

Reviewer vyqg may increase their score from 4 to 6, as the major concerns are adequately addressed in the rebuttal.

Reviewer 22z7 is likely to maintain a score of 4, given that concerns about overlap with existing techniques remain unresolved.

Reviewer C1zb may keep their original score of 2 or increase it to 4. Although some concerns are addressed, this reviewer remains somewhat skeptical of the reported comparisons, which appear to be sensitive to implementation details.

---

### Decision · Program_Chairs · 2026-01-26

Reject